# IDEA: ENHANCING THE RULE LEARNING ABILITY OF LARGE LANGUAGE MODEL AGENT THROUGH INDUCTION, DEDUCTION, AND ABDUCTION

## ABSTRACT

While large language models (LLMs) have been thoroughly evaluated for deductive and inductive reasoning, their proficiency in abductive reasoning and holistic rule learning in interactive environments remains less explored. We introduce RULEARN, a novel benchmark specifically designed to assess the rule-learning abilities of LLM agents in interactive settings. In RULEARN, agents strategically interact with simulated environments to gather observations, discern patterns, and solve complex problems. To enhance the rule-learning capabilities for LLM agents, we propose IDEA, a novel reasoning framework that integrates the process of **I**nduction, **DE**duction, and **A**bduction. The IDEA agent generates initial hypotheses from limited observations through abduction, devises plans to validate these hypotheses or leverages them to solve problems via deduction, and refines previous hypotheses using patterns identified from new observations through induction, dynamically establishing and applying rules that mimic human rule-learning behaviors. Our evaluation of the IDEA framework, which involves five representative LLMs, demonstrates significant improvements over the baseline. Furthermore, within this framework, our comparison with 50 human participants reveals notable discrepancies in rule-learning behaviors. LLM agents tend to generate plausible initial hypotheses but struggle to refine them through interaction. Conversely, humans, despite sometimes overlooking initial details, excel at incorporating feedback and continuously improving their hypotheses. We believe our benchmark, RULEARN, will serve as a valuable and challenging resource, and that the IDEA framework will provide crucial insights for the development of LLM agents capable of human-like rule learning in real-world scenarios. We will release our code and data upon acceptance of the paper.

## 1 INTRODUCTION

One major pillar of human intelligence is the ability to discern rules and apply them. We identify patterns, formulate hypotheses, and refine them by interacting with the environment. This exploratory process traditionally involves three stages: abduction, deduction, and induction. According to Charles Peirce's definition (Frankfurt, 1958; Peirce, 1974), the rule-learning loop typically begins with an explanatory hypothesis that arises from **abduction**. This is followed by iterative experiments guided by the hypothesis, known as **deduction**, which leads to the modification and refinement of the hypothesis through **induction** (As shown in Figure 1). Whether the hypothesis emerges from rigorous logical reasoning or a more creative process is debated, but the cyclic nature of this reasoning is pivotal in advancing our under-

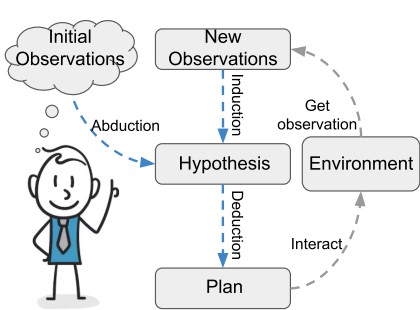

Figure 1: The reasoning cycle of rule learning encompasses abduction, deduction, and induction.

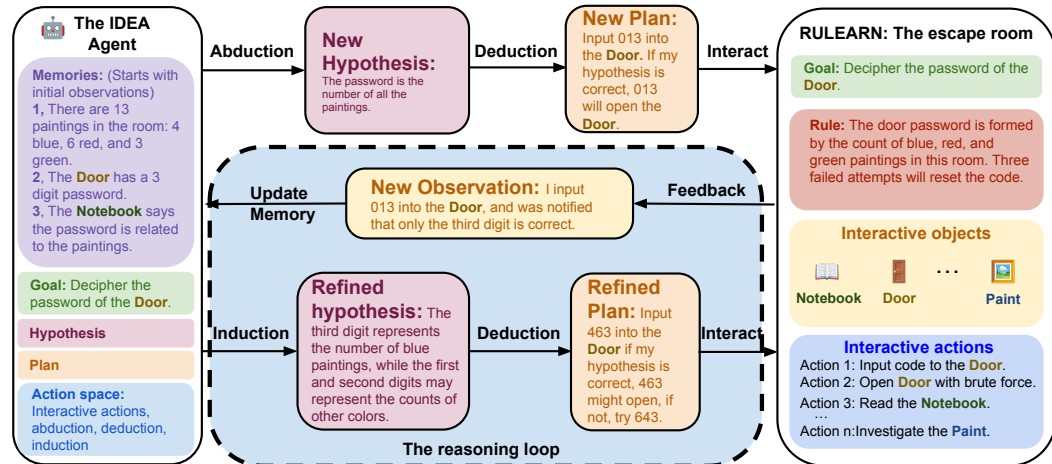

Figure 2: A simplified puzzle in the RULEARN benchmark and the IDEA agent's workflow (in real puzzles, agents have fewer initial observations and more complex rules). The agent generates an initial hypothesis through abduction, develops an exploration plan via deduction, and refines its hypothesis using induction. For example, the IDEA agent first hypothesizes that the password is the number of the blue paintings, tests this by entering code 003, and adjusts its strategy based on the results.

standing and problem-solving capabilities in various disciplines. This establishes the foundation of how humans learn the rules of the world and apply them to effect change.

For such cyclic reasoning to function effectively, it requires an interactive foundation. In the real world, the observations we reason from are gathered through interaction. Then, through alternating cycles of refining hypothesis and interaction, we iteratively collect evidence and refine our hypotheses, drawing ever closer to the truth. The dynamic interplay between interaction and different stages of the reasoning loop within an interactive environment is the key to humans' reasoning effectiveness. However, recent studies focusing on reasoning often lack this dynamic and interdependent nature, as abduction, induction, and deduction are tested one at a time in non-interactive environments (Bowen et al., 2024; Wang et al., 2023; Saparov et al., 2024; Liu et al., 2024).

To address this gap, we introduce RULEARN, a benchmark designed to evaluate the rule-learning abilities of LLM agents in interactive environments. RULEARN consists of puzzles with hidden rules set in a text-based environment, where agents begin exploration without prior knowledge of these rules. The agent's objective is to solve the puzzle by interacting with the environment. At each step, the agent selects an action from a predefined set, executes it, and gathers observations to gain insights into the environment. Successfully solving the puzzles requires the agent to strategically select actions, efficiently gather pattern-revealing observations, and accurately reason from them to infer the hidden rules. RULEARN presents substantial challenges, as agents must rely entirely on observations generated by their chosen actions to discern rules. If these observations fail to reveal a clear pattern, the agent is likely to fail.

We design three types of environments within RULEARN to evaluate the rule-learning ability in different scenarios: (1) **The Function Operator:** The agent is tasked with determining the coefficients of mathematical functions defined by hidden expressions. Agents can assign various values to the input variables and observe the outputs, using this information to hypothesize the function's form. The challenge lies in efficiently selecting input values that reveal the underlying structure and accurately computing the coefficients based on limited observations. (2) **The Escape Room:** The agent is tasked with deciphering the passcode to exit an escape room. A hidden rule determines how the objects in the room infer the passcode digits. Agents interact with these objects to gather clues and input passcodes into the door. The door provides hints indicating which digits are correct. Based on these hints, agents formulate hypotheses and infer the relationship between the objects and the passcode. (3) **The Reactor:** The agent is tasked with synthesizing target strings using a reactor with a hidden string-combining rule. Agents need to experiment with different inputs and analyze outputs to deduce the reactor's transformation rule and achieve the desired outcome. Detailed puzzle definitions are provided in § 3.

To tackle the challenge in RULEARN, we introduce IDEA, a novel reasoning framework that integrates the process of **I**nduction, **DE**duction, and **A**bduction. The IDEA agent employs these reasoning processes iteratively to explore the environments, learn rules, and achieve goals. In the **abduction** phase, the IDEA agent generates an initial hypothesis from limited observations. During the **deduction** phase, the IDEA agent creates and executes plans to meet objectives or test its hypothesis. In the **induction** phase, the IDEA agent refines its hypothesis based on new observations, enhancing their accuracy and robustness. This iterative cycle enables the LLM agent to continually improve the learned rules through environmental feedback. An overview of how the IDEA agent solves puzzles in RULEARN is shown in Figure 2.

We evaluat the IDEA framework using five popular LLMs—GPT-3.5-Turbo, GPT-4o, Gemma-7B, Llama3-8B and Llama3-70B. The experimental results demonstrate that IDEA significantly improves rule-learning proficiency and puzzle-solving performance of the LLM agent within the RULEARN benchmark. The IDEA agent increases success rates by approximately 10% compared to the baseline LLM agent. Specifically, the action of the IDEA agent aims to validate or leverage current hypothesis, providing important guidance in an environment with hidden rules. Without the guidance of the hypothesis, the LLM agent will always choose the most direct actions attempting to quickly solve the puzzle. Such actions, taken without understanding the hidden rules, are akin to navigating without a compass—efforts are made, but they often lead to ineffective or misguided outcomes. Based on our experiments, the IDEA agent achieves 47.6% fewer repeated actions, yielding more diverse observations and finally solving the puzzles with a better understanding of the hidden rule. These improvements suggest that IDEA enables LLMs to better adapt and learn rules in unfamiliar environments, narrowing the gap between human and AI rule-learning abilities.

Within the IDEA framework, we conduct a study to compare the rule-learning behaviors and performances between LLMs and 50 human participants solving puzzles. Our findings reveal that, despite the improvements brought by the IDEA framework, LLMs still face the following challenges when compared to human participants: (1) LLM agents continue to struggle with efficiently exploring unfamiliar environments. The new observations gathered usually do not provide enough information to help reveal the underlying rules, moreover, LLM agents tend to repeat previous actions that do not yield new information. (2) LLMs fall short in deducting valid plans to verify current hypotheses and guide future exploration. (3) LLMs fall short to correct previous hypotheses when they contradict new observations and are less capable of refining a hypothesis to make it more robust and encompassing all observations. These findings provide valuable insights for the development of LLM agents capable of human-like rule learning in real-world scenarios.

## 2 RELATED WORKS

### 2.1 LLM AGENT

Agents driven by large language models equipped with memory, tool use, planning, and reasoning capabilities have shown significant advancements in understanding complex tasks and achieving notable performance (Wang et al., 2024b). Techniques like memory summarization (Chen et al., 2023; Zhou et al., 2023; Wang et al., 2024a) and retrieval (Andreas, 2022; Park et al., 2023; Zhong et al., 2023) assist LLM agents in filtering relevant information based on the current context (Zhang et al., 2024). Additionally, the integration of tool use enables LLM agents to perform better in knowledge-grounded tasks. For instance, some studies (Nakano et al., 2022; Lu et al., 2023; Shi et al., 2023a) have enabled LLM agents to access up-to-date information from external databases to provide correct answers, while others (Schick et al., 2023) have facilitated the use of various tools like calculators to enhance solution accuracy. Planning modules (Yuan et al., 2023; Shen et al., 2023) allow these agents to decompose complex tasks into simpler components, achieving significant improvements. Moreover, employing search algorithms (Yao et al., 2023; Besta et al., 2024) helps LLM agents reach solutions in fewer steps. Building on this foundation, we are interested in whether LLM agents can derive new knowledge–learning rules through interacting with the environment, rather than merely answering questions. Recent studies have begun investigating different types of reasoning in LLMs, including abduction, deduction, and induction (Bowen et al., 2024; Wang et al., 2023; Saparov et al., 2024; Cheng et al., 2024; Yang et al., 2024). However, existing research mostly focuses on isolated aspects of the rule-learning process. Consequently, the compre-

hensive rule-learning abilities of LLMs in interactive environments remain largely underexplored, a topic that will be further examined in Section 2.2.

## 2.2 RULE LEARNING

Rule learning—involving evidence gathering, pattern identification, hypothesis formulation, and validation—is gaining focus in recent research. Yang et al. (2023) and Zhu et al. (2024) improved question-answering systems by formulating optimal response rules, while Shi et al. (2023b) advanced event prediction by enabling LLMs to hypothesize causes of previous events. Some works also evaluate LLMs rule-learning abilities (Liu et al., 2024). However, existing studies adopt static, non-interactive settings for rule learning, where LLMs are provided with complete information to formulate rules. This approach has two significant limitations: it contrasts with real-world rule learning, which involves actively and strategically gathering evidence, and it restricts agents from conducting experiments to validate the rules they learn. In real-world scenarios, the process of gathering observations, validating hypotheses, and refining hypotheses is iterative. These simplified, static settings fail to adequately assess an agent's capabilities in real, dynamic environments. Even in studies with interactive environments (Xu et al., 2024; Montes et al., 2022), information gathering, rule generation, and application are treated as independent objectives or are incomplete. In reality, these processes are dynamic and interdependent: generated hypotheses guide actions, and new observations refine those hypotheses. In contrast, RULEARN provides a realistic setting that allows LLMs to engage in dynamic, interdependent rule-learning processes, closely mirroring real-world scenarios. To effectively navigate and learn within this environment, IDEA equips LLM agents with the ability to emulate human reasoning by managing these interdependent processes.

## 3 THE RULEARN BENCHMARK

We develop three puzzle sets, the **Function Operator**, the **Escape Room**, and the **Reactor**, each comprising 20 puzzles associated with environments of varying complexity.

**The Function Operator.** This puzzle type simulates scenarios where systemic theories or established knowledge (in this case, mathematics) is applicable. The agent interacts with a set of univariate multi-term equations involving integer parameters from [0,9] and elementary functions of the variable $x$, selected from $f(x) \in \{x^0, x^1, x^2, \sin(x), \frac{1}{x}, |x|, -x\}$. The agent is provided with the number of functions, the presence of specific parameters in each function (the exact numerical values of these parameters are unknown and represented by letters), and the types of elementary functions involved in the current puzzle. The goal of the agent is to deduce the values of these parameters. For example, in one puzzle, the ground truth is $\mathbf{F_1}(x) = a\sin(x) + b \times \frac{1}{x}$, $\mathbf{F_2}(x) = ax^2$ where $a = 3$ and $b = 2$. The agent knows the following information: There are three elementary functions in this puzzle $\{\sin(x), \frac{1}{x}, x^2\}$, there are two functions $\mathbf{F_1}(x)$ and $\mathbf{F_2}(x)$, $\mathbf{F_1}(x)$ has 2 terms and parameters $a, b$ in it, and $\mathbf{F_2}(x)$ has 1 term and one parameter $a$. To solve the puzzle, the agent must interact with the environment through a defined action space: selecting a function and assigning values to $x$, then observing the resultant output. For example, assigning values 1 and 2 to $\mathbf{F_2}$ reveals a quadratic increase in output, indicating the presence of $x^2$ in $\mathbf{F_2}$. Similarly, assigning a value of 1 to $\mathbf{F_1}$ results in a floating-point output, rather than an integer, suggesting the inclusion of trigonometric components, confirming that $\sin(x)$ is a component of $\mathbf{F_1}$. Each puzzle features a different number of functions, unknown parameters, and elementary functions in use.

**The Escape Room.** This environment simulates scenarios where no established knowledge is applicable. We create a fictitious environment: an agent is in an escape room–an art gallery, tasked with deciphering a password to unlock a code-secured door and escape. The password is a 3-digit number where each digit represents the count of paintings of a specific **type**—watercolor, oil, or acrylic—that share a given **color**. The agent can explore the environment and investigate paintings, receiving brief descriptions such as *This is an acrylic painting of a green jungle*, providing information about the painting's type and color. The agent only knows that the password is a 3-digit number and receives a hint to focus on a specific color relevant to the password. After forming an initial hypothesis and inputting the password into the door, the door provides hints indicating which digits are correct. Based on these hints, the agent refines its hypotheses and infers the relationship between the painting attributes and the passcode. To prevent the problem from being solved merely by brute force, the specific color associated with the password changes after every three failed attempts. Each

Table 1: The reacting rules in the Reactor Puzzle. All letters are functionally equivalent and exhibit no special behaviors. Identical symbols represent the same letter, while different symbols denote different letters. Each puzzle operates under one specific rule. The Middle Insertion rule inserts the shorter string into the longer string; if the length of the longer string is odd, the shorter string is inserted just to the right of the center. If both strings are of equal length, the second string is inserted into the middle of the first string. The Prefix Replacement rule retains the prefix of the longer string and concatenates it with the shorter string, dropping the tail of the longer string results in two output strings. There are two special cases where the strings are simply concatenated in order.

| Rule Description | Example Reaction 1 | Example Reaction 2 | Special Case 1 | Special Case 2 |
|---|---|---|---|---|
| Simple Concatenation | AB + C = ABC | AB + CDE = ABCDE | — | — |
| Reverse Concatenation | AB + C = CAB | AB + CDE = CDEAB | — | — |
| Middle Insertion | AB + C = ACB | AB + CDE = CDABE | A + B = AB | — |
| Prefix Replacement | AB + C = AC + B | AB + CDE = CAB + DE | AB + CD = ABCD | AA + A = AAA |

puzzle presents a varying number of paintings; paintings in the same room as the agent are directly visible, while those in different rooms are not, requiring the agent to take actions such as moving to another gallery to reveal all the paintings.

**The Reactor.** This environment type also simulates scenarios where no established knowledge is applicable. However, unlike in the Escape Room where agents only input passwords to test hypotheses, in the Reactor, agents are challenged to design more fine-grained experiments to uncover hidden rules. Specifically, the agent's task is to synthesize target strings using a reactor governed by a hidden string-combining rule. These strings are represented by sequences of alphabetic letters, such as *A*, *B*, *AABB*, and *CAB*. The reactor permits the agent to input two strings, initiating a reaction that produces a new string for use in subsequent experiments. The agent's objective is to decipher the specific rules that govern string synthesis by methodically testing different string combinations, with the ultimate goal of synthesizing the target string using the discovered rules. We have designed four types of rules, detailed in Table 1.

The detailed statistics for each puzzle type and example puzzles are provided in Appendix A.2 and A.5, respectively. Note that the RULEARN benchmark is designed to simulate real-world interactions and is adaptable to complex text environments with diverse rules. To achieve this, we do not restrict the representation of rules to a specific formal language, as doing so would limit realism and complexity. Moreover, predefining the rule format would provide agents with prior knowledge, contradicting real-world conditions. Therefore, we enable the LLM agents to use natural language to describe rules in RULEARN, promoting generalizability and future extensions of the benchmark.

## 4 THE IDEA AGENT

We introduce IDEA, a novel reasoning framework that integrates the process of **I**nduction, **DE**duction, and **A**bduction to learn rules in interactive environments. The IDEA agent consists of the following components: Goal(G), Action Space($\mathbb{A}$), Memory($\mathbb{M}$), Hypothesis(H), and Plan(P), which are elaborated in Table 2.

Upon beginning to explore a puzzle, we initialize the agent memory with an initial observation of the environment. The agent's goal is initialized with the objective of the puzzle, e.g., *synthesize a target string* for a Reactor puzzle. The agent's action space is initialized as the set of interactive actions defined by the puzzle, such as *choosing two strings and running the Reactor*, as well as establish the initial hypothesis (abductive action), devises a plan to validate or leverage hypothesis (deductive action), and refining the current hypothesis (inductive action).

The IDEA agent begins with an abductive action to generate an initial hypothesis, followed by a deduction step to create a new plan. Based on this plan, the agent interacts with the environment. Upon receiving feedback from the environment as a new observation, the agent may take an inductive action to refine the hypothesis or perform another interaction with the environment. Deductive action is invoked to adjust the plan every time the hypothesis changes. This reasoning loop continues until the puzzle is solved or a maximum number of steps is reached. After each step, the results are appended to the agent's memory, including interaction outcomes and any modifications to the hypothesis or plan. We provide a simplified algorithm demonstrating how the IDEA agent

**Algorithm 1** IDEA Agent Rule-learning Loop

1: **procedure** RULELEARNINGLOOP
2:    Initialize Goal(G), Action Space($\mathbb{A}$)
3:    Memory($\mathbb{M}$) $\leftarrow$ Initial observations
4:    #step $\leftarrow$ 0
5:    Hypothesis(H) $\leftarrow$ **Abduct**(G, $\mathbb{A}$, $\mathbb{M}$)
6:    Plan(P) $\leftarrow$ **Deduct**(H, G, $\mathbb{M}$, $\mathbb{A}$)
7:    $\mathbb{M}$.add("New hypothesis and plan", H, P)
8:    **while G** not achieved and #step $\leq$ max_step **do**
9:       a $\leftarrow$ select_action(G, H, P, $\mathbb{M}$, $\mathbb{A}$)
10:       **if a** is interactive action **then**
11:          result $\leftarrow$ execute_action(a, G, H, P, $\mathbb{M}$)
12:          $\mathbb{M}$.add(result)
13:          #step $\leftarrow$ #step $+ 1$
14:       **else if a** is inductive action **then**
15:          H $\leftarrow$ **Induct**(a, G, $\mathbb{M}$, H, P)
16:          P $\leftarrow$ **Deduct**(H, G, $\mathbb{M}$, $\mathbb{A}$)
17:          $\mathbb{M}$.add("Refined hypothesis and plan", H, P)
18:       **end if**
19:    **end while**
20: **end procedure**

| The IDEA Agent Component | Definition |
|---|---|
| Goal(G) | Goal of the agent in the current puzzle. |
| Action Space($\mathbb{A}$) | Set of actions the agent can take, including abductive action, deductive action, inductive action, as well as the set of interactive actions defined by the puzzle. |
| Memory($\mathbb{M}$) | Set of natural language strings to record all interaction results till the current step. |
| Plan(P) | Generated plans to guide future actions. |

Table 2: Components of the IDEA agent.

operates in Algorithm 1. Specifically, at each step, we prompt the LLM to reflect on the information recorded in the IDEA agent's components to make decisions and take actions. We employ the chain-of-thought (COT) reasoning (Wei et al., 2022) for all prompts. Detailed prompts for each type of action are available in Appendix A.4. More detailed implementation of the agent can be found in Appendix A.3.1. Similar to real-life scenarios, when agents solve tasks in RULEARN puzzles, they do not know the outcomes in advance. Consequently, it is challenging to decide when to refine or change their hypothesis and plans, as well as what interactive actions to take to gather pattern-revealing observations. A detailed example of the IDEA agent solving the Reactor Puzzle is provided in Figure 3.

### 4.1 EXPERIMENT SETTINGS

To evaluate the effectiveness of IDEA, we respectively initialize it with three popular open-source LLMs, including Gemma-7B (Team et al., 2024), Llama3-8B, and Llama3-70B (Dubey et al., 2024), and two closed-source LLMs, GPT-3.5-Turbo (Dubey et al., 2024) and GPT-4o [1]. We compare the IDEA agent against the following two variants:

- **Baseline Agent**: The Baseline agent lacks the reasoning loop of abduction, deduction, and induction and does not generate hypotheses or plans. At each step, it selects an interactive action solely based on its current memories and the goal.
- **Oracle-rule Agent**: Even if the agent could successfully learn the correct rule, applying the learned rule to solve the puzzle is non-trivial. The Oracle-rule agent serves as a special control group to establish the Oracle performance with the ground-truth rule provided at the beginning. Specifically: 1) For the Function Operator puzzles, agents are given the exact forms of the functions. Their task is to derive the values of the coefficients. 2) For the Escape Room puzzles, agents are provided with how the password is constructed from the objects in the room. Their task is to derive the password using the provided rule. 3) For the Reactor puzzles, the reaction rule is given to the agents in natural language accompanied with examples. The agents only need to synthesize the target strings using the provided rules.

Each variant is evaluated on all three puzzle types (totaling 60 puzzles), with five trials per puzzle, resulting in 300 tests per variant. In our initial experiments, we found that using a higher temperature enhanced the agent's creativity, allowing it to generate more diverse and higher-quality hypotheses

---
[1]https://openai.com/index/gpt-4o-system-card/

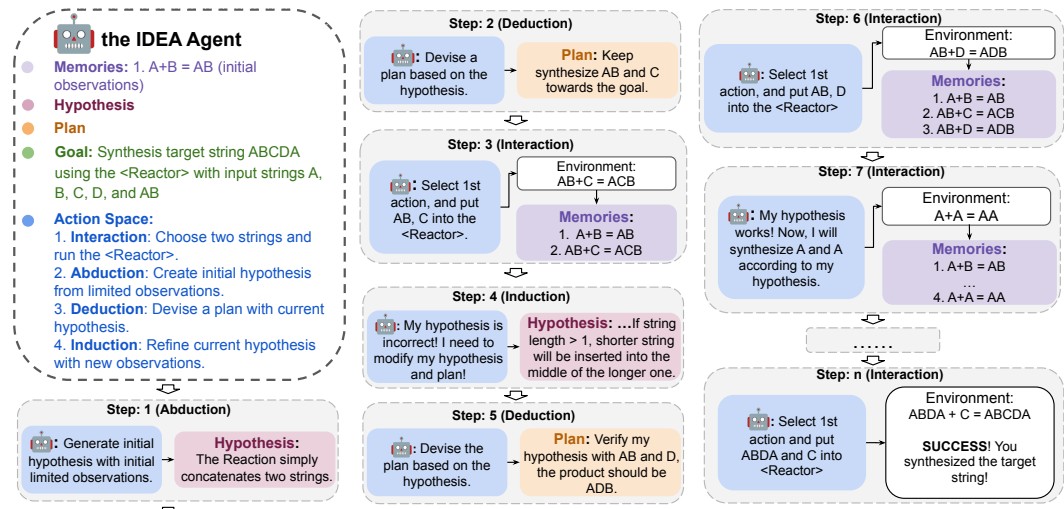

Figure 3: An example of the IDEA agent solving a Reactor puzzle. At each step, the agent must choose whether to interact with the environment or adjust its hypothesis and plan based on current observations. If observed facts contradict the existing hypothesis, the agent is expected to refine its hypothesis. The refined hypothesis and plan will then guide subsequent exploration.

that aided in discovering the underlying rules. Therefore, we set the temperature to $1.0$. We apply the COT for the prompting of each step for all agents. Additionally, since we observed that the success rate does not improve after 15 interactive steps, we set the maximum interaction step count to 15. The agent is considered to have failed a puzzle if it doesn't solve it within 15 interactive steps.

## 4.2 HUMAN PARTICIPANTS

To compare the behavior and performance of humans with that of LLMs in the abduction, deduction, and induction reasoning processes, we recruited 50 participants and randomly selected 30 puzzles (10 from each of the three puzzle types), assigning each participant three puzzles—one from each type. Each puzzle was attempted by five different individuals, ensuring no prior knowledge of the underlying rules and maintaining an unbiased sample across puzzle types. Details about Institutional Review Board (IRB) approval and participant recruitment are provided in §7. The human participants are instructed to operate under the same reasoning procedure as IDEA. Failure to solve a puzzle within 15 steps resulted in marking the attempt as unsuccessful, ensuring a fair comparison.

## 4.3 MAIN RESULTS

We calculated the Wilson Confidence Interval (CI) (Brown et al., 2001) for puzzle solving success rate across different variants. The detailed results are displayed in Table 3.

For the Oracle-rule agent, GPT-4o exhibits superior deductive capabilities, outperforming other models with an approximate 50% success rate across all three puzzle types. Specifically, in the Escape Room puzzles, agents achieve up to an 80% success rate by simply following the provided rule to count the number of paintings of specific colors and types. However, in the Function Operator and Reactor puzzles, merely knowing the rule is not sufficient for success, applying the rule to solve the puzzle is more challenging. Agents must also devise effective plans to apply these rules to reach a solution. This additional planning requirement results in a comparatively lower success rate in these puzzles.

For the Baseline agent, where they are not provided with the underlying rules and solely rely on historical observations to make interactive actions, GPT-4o continues to lead in performance. Notably, Llama3-70B narrows the performance gap, especially in the Function Operator puzzles. Across models, the success rates drop by about half compared to the Oracle-rule agent. This significant decrease highlights the challenge of rule learning and indicates that current LLM agents struggle to learn rules in unfamiliar environments without explicit guidance.

Table 3: Puzzle Success Rate (95% CI). The table displays the success rate for each setting, with each pair of numbers in parentheses representing the lower and upper bounds of the 95% confidence interval for the success rate. The human results are based on a randomly selected 50% subsample of the puzzles. We can see that across all LLMs, IDEA achieves consistent significant improvements except for the Reactor puzzles. We use **Boldface** to highlight the performance comparisons between Baseline and IDEA Agents with GPT-4o.

| Setup | LLMs | All Types (%) | Function Operator (%) | Escape Room (%) | Reactor (%) |
|---|---|---|---|---|---|
| **Oracle-rule Agent** | Gemma-7B | (0.71, 3.84) | (0.0, 3.7) | (1.57, 9.84) | (0.18, 5.45) |
| | Llama3-8B | (7.09, 13.92) | (2.78, 12.48) | (12.51, 27.78) | (2.15, 11.18) |
| | Llama3-70B | (33.33, 44.29) | (28.18, 46.78) | (48.21, 67.2) | (14.17, 29.98) |
| | GPT-3.5-Turbo | (3.83, 9.28) | (0.55, 7.0) | (4.11, 15.0) | (4.11, 15.0) |
| | GPT-4o | (47.35, 58.57) | (35.61, 54.76) | (72.22, 87.49) | (24.56, 42.69) |
| **Baseline Agent** | Gemma-7B | (0.34, 2.9) | (0.0, 3.7) | (0.18, 5.45) | (0.55, 7.0) |
| | Llama3-8B | (2.55, 7.27) | (0.55, 7.0) | (4.81, 16.23) | (0.55, 7.0) |
| | Llama3-70B | (18.6, 28.09) | (22.78, 40.63) | (12.51, 27.78) | (12.51, 27.78) |
| | GPT-3.5-Turbo | (3.57, 8.89) | (5.52, 17.44) | (1.03, 8.45) | (1.57, 9.84) |
| | GPT-4o | **(19.21, 28.79)** | **(18.6, 28.09)** | **(14.17, 29.98)** | **(14.17, 29.98)** |
| **IDEA Agent (Ours)** | Gemma-7B | (0.06, 1.86) | (0.18, 5.45) | (0.0, 3.7) | (0.0, 3.7) |
| | Llama3-8B | (1.36, 5.17) | (0.18, 5.45) | (3.43, 13.75) | (0.0, 3.7) |
| | Llama3-70B | (25.41, 35.76) | (34.67, 53.77) | (21.89, 39.58) | (10.89, 25.55) |
| | GPT-3.5-Turbo | (10.53, 18.38) | (22.78, 40.63) | (2.78, 12.48) | (2.15, 11.18) |
| | GPT-4o | **(30.14, 40.9)** | **(30.94, 49.8)** | **(32.8, 51.79)** | **(16.69, 33.23)** |
| | Human | (55.38, 70.62) | (52.15, 77.56) | (42.31, 68.84) | (54.19, 79.24) |

**The IDEA Agent significantly boosts success rates.** Our proposed IDEA framework leads to approximate 10% increases in success rates for Llama3-70B, GPT-3.5-Turbo, and GPT-4o compared to the Baseline agent. This improvement demonstrates that incorporating a reasoning loop of abduction, deduction, and induction substantially enhances the LLM rule-learning performance in unfamiliar environments. IDEA enables the LLMs to generate hypotheses, plan actions, and refine their understanding based on new observations, which is crucial for rule learning. However, smaller models like Llama3-8B and Gemma-7B experience a slight decline in performance when applying IDEA. This decline is likely due to the increased complexity and context management demands introduced by the IDEA framework, which pose challenges for small models with limited capacity.

**LLM agents still fall far behind humans.** In the Escape Room puzzle, where the primary challenge is to discover the rule, the Oracle-rule agents excel because once the rule is identified, applying the rule is simple. However, in other types of puzzles, human participants significantly outperform all LLM agents, including the Oracle-rule agents, even without knowing the rules beforehand. This notable disparity highlights that LLM agents lag substantially behind humans, not only in learning new rules but also in effectively applying known rules to solve problems.

## 4.4 ANALYSIS

**IDEA boosts puzzle-solving speed.**
Figure 4 illustrates the cumulative number of puzzles solved at each interaction step for the Baseline agent, the IDEA agent, and human participants. The slopes of the lines represent the rate at which puzzles are solved per step. Compared to the Baseline agent, the IDEA agent exhibits a steeper slope, indicating that the integration of abductive, deductive, and inductive reasoning enhances the agent's efficiency in exploring the environment and learning the underlying rules, especially during the early stages.

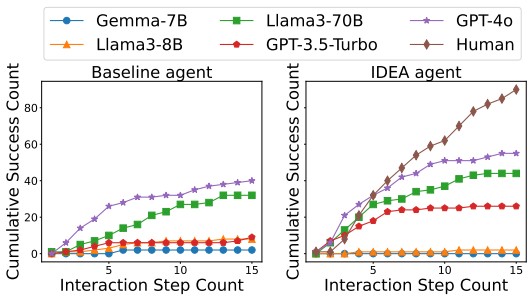

Figure 4: Comparison of the cumulative number of puzzles solved at each interaction step. The IDEA agent significantly increases the solving speed compared to the Baseline agent.

When focusing on human participants, we observe that they solve fewer puzzles in the initial steps compared to the LLM agents. However, as interactions continue, the number of puzzles solved by humans increases rapidly, eventually surpassing that of all LLM agents. In contrast, LLM agents

solved 88.76% of the puzzles within the first 10 steps. Beyond this point, additional interactions contribute less to their success rate. This pattern suggests that humans have a superior ability to learn continuously from interactive environments, effectively improving their performance over time. If we did not limit the puzzles to 15 steps, we anticipate that the success rate of human participants would be even higher. We will explore this observation further in Section 5.

**IDEA reduces repetitive actions.** LLM agents frequently repeat previous actions instead of exploring new ones. This behavior is highly inefficient in our controlled puzzle environments, where each interaction yields deterministic results, and repeating the same action generally does not provide new information. An exception is the Escape Room puzzle, where entering the same password could result in different outcomes after the rule changes following three failed attempts.

We calculate the average number of repeated actions performed while solving each puzzle, with detailed statistics presented in Table 7 in Appendix A.2. We observe that most LLMs commonly repeat actions in the Baseline agent. Although models like GPT-4o and Llama3-70B exhibit this issue to a lesser degree, even these advanced models sometimes repeat actions that fail to provide further insights. The IDEA agent effectively reduces this tendency by explicitly generating plans during the deduction phase. By outlining a clear plan, the IDEA agent can better assess whether the current observations are sufficient or if further specific evidence is needed to reveal the underlying rule. For example, in the Escape Room puzzle, the IDEA agent avoids unnecessary attempts at entering passwords when the evidence gathered is sufficient to determine the correct code (see Figure 11 in Appendix A.4).

**IDEA relies on the reasoning ability of underlying LLMs.** The effectiveness of IDEA depends on the underlying LLMs' ability to reason from hypotheses and observations. Particularly, if an agent generates a false hypothesis and fails to properly refine it, being guided by this incorrect hypothesis can lead the agent to perform even worse than the baseline. During our experiments, we observed that current LLMs tend to hallucinate, especially in the Escape Room puzzles and more severely in the Reactor puzzles. This results in smaller performance improvements compared to those seen with the Function Operator puzzles. This is likely because such fictitious scenarios are not extensively represented in LLM training data. Moreover, LLMs struggle to recognize letter-level patterns, and their reasoning capabilities still require significant enhancement. Examples of hallucination can be seen in Appendix A.4.4).

## 5 HUMAN EVALUATION

We employ three computer science graduate students (our co-authors) to evaluate all the hypotheses and plans generated by both the IDEA agent and human participants during the abduction, deduction, and induction stages within a randomly selected 50% subsample of the puzzles.

**Abduction stage.** During the abduction stage, agents formulate an initial hypothesis based on initial observations. Since some puzzles are simple, agents can potentially guess the ground truth rule at this stage. Figure 5(a) shows that LLMs such as GPT-4o have approximately a 30% success rate in correctly identifying the rule during abduction. Notably, humans exhibit the lowest success rate in this stage. We believe this is because humans often lack the patience to thoroughly review all textual information initially and may overlook certain details. In contrast, LLMs meticulously process every word of the prompt and leverage their comprehensive pre-training data to generate plausible hypotheses. Additionally, humans tend not to formalize hypotheses when operating under uncertainty.

**Deduction stage.** After establishing an initial hypothesis—or each time the agent refines a hypothesis—the agent derives a plan to either validate this hypothesis or use it to solve the puzzle. As shown in Figure 5(b), humans generally outperform LLMs in creating high-quality plans. However, in the Function Operator puzzles, Llama3-70B and GPT-4o perform better than humans, suggesting that these models may have superior mathematical intuition compared to our human participants, who were undergraduate students from computer science-related majors.

**Induction stage.** Figure 5(c) illustrates the proportion of inductions where the refined hypothesis improved over the previous one, which we define as the effective induction rate. Humans outperform LLMs by a significant margin in this process, with 40% of their refined hypotheses being better than the previous ones. Surprisingly, GPT-4o performs worse than Llama3-70B and GPT-3.5-Turbo.

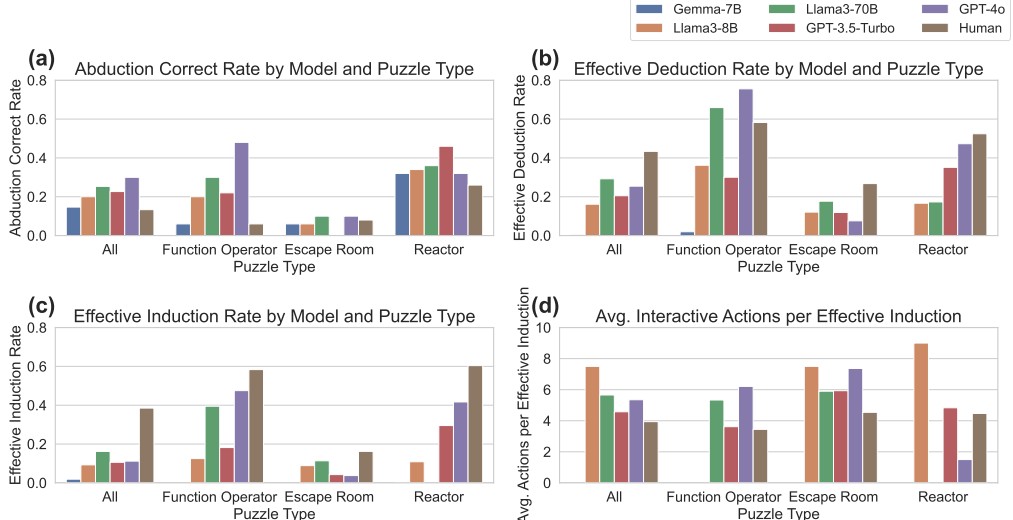

Figure 5: Human Evaluation Results. Bars represent measured values per model and puzzle type; the absence of a bar indicates zero or unavailable data. Plot (a): Abduction Correct Rate—the frequency of correctly guessing the rule during abduction. Plot (b): Effective Deduction Rate—the rate at which deduction plans effectively validate hypotheses or solve puzzles. Plot (c): Effective Induction Rate—the proportion of inductions where the refined hypothesis improved over the previous one. Plot (d): Average Actions per Effective Induction—the average number of interactive actions needed for an effective induction.

We found that GPT-4o tends to hallucinate, especially in Escape Room puzzles, which negatively impacts its overall effective induction rate. In contrast, Llama3-70B almost never engages in induction activities within Reactor puzzles, as it fails to recognize when its hypothesized rules contradict its observations (see Appendix A.4.4). Despite these issues, both Llama3-70B and GPT-4o exhibit performance closer to humans during the induction process compared to other models in Function Operator puzzles. This relative success is believed to be due to the extensive training of LLMs with mathematical content, which ensures stable and superior performance in these types of puzzles. In contrast, Escape Room and Reactor puzzles, which are less likely to appear extensively in training datasets, see poorer and more variable performance from the models.

**Average interactions needed for effective induction.** Figure 5(d) shows that humans require fewer interactions—approximately four on average—to effectively refine their hypotheses, compared to LLMs. While LLMs can process initial information thoroughly and generate plausible hypotheses, they face challenges in refining these hypotheses based on new observations during interaction with the environment (see Figure 7 in Appendix A.1). This limitation suggests that LLMs may struggle to learn from new observations and incorporate feedback to continuously improve their hypotheses and problem-solving strategies. This gap may become more pronounced when agents are faced with larger action spaces and environments governed by more complex rules.

## 6 CONCLUSION

In this work, we introduce RULEARN, a benchmark designed to evaluate LLM agents' rule-learning abilities in interactive environments. We propose IDEA, an agent framework that mimics human reasoning through abduction, deduction, and induction. Comprehensive experiments involving five prominent LLMs and human participants reveal that while IDEA significantly improves the rule learning ability of LLM agents, there is still a large gap between LLM and humans particularly in refining hypotheses and adapting strategies. Despite these advancements from the IDEA framework, LLMs still face challenges in generating valid hypotheses and avoiding repetitive actions in complex scenarios. Our findings underscore the need for further development of LLMs that can emulate human cognitive processes more effectively in explorations of novel environments. RULEARN provides a foundational resource for future research aimed at closing these gaps.

## 7 ETHICS STATEMENT

Our work aims to benefit the broader research community by introducing RULEARN, a benchmark for evaluating the rule-learning abilities of LLM agents and proposing the IDEA agent framework. All data in RULEARN contains no personal or sensitive information, ensuring respect for privacy and ethical standards. This project is approved by our Institutional Review Board (IRB). Human participants are recruited through emails from our university's computer science and engineering department. All participants were adults over 18 years old and provided informed consent. The data collected from these participants were de-identified and consented for release for research purposes. Participants were compensated $15 each for one hour of their time. We ensured that all content presented during evaluations was free from offensive or inappropriate material. For human evaluations of all the hypotheses and plans generated by LLM agents and human participants, three computer science graduate students (our co-authors) conducted the evaluation. We are committed to the ethical use of our benchmark and agent framework, and upon acceptance of this paper, we will release our code and data to encourage open collaboration and advancement in the field.

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

# A APPENDIX

## A.1 FIGURES

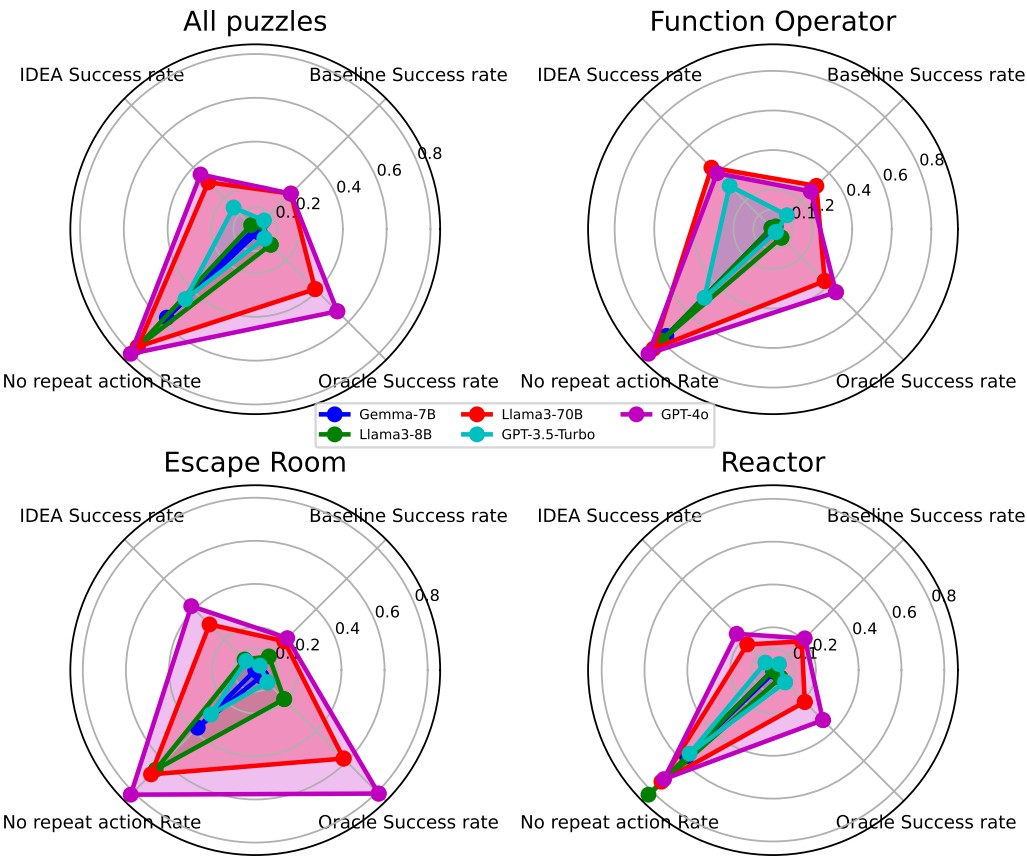

Figure 6: The scaled performance radar plot shows varying performances across different puzzle types. GPT-4o leads, followed by Llama 70B, GPT-3.5, Llama 8B, and Gemma 7B.

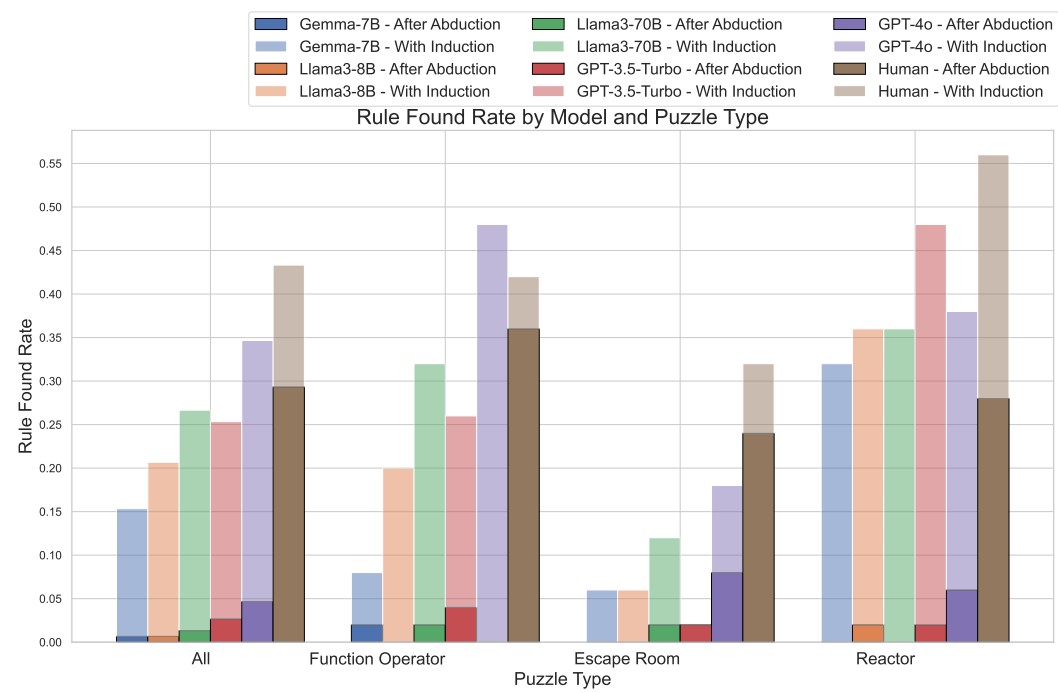

Figure 7: Although agents continuously refine their hypotheses toward the ground truth rule, identifying the exact rule remains a challenging task. According to our evaluation, humans have a 43.3% success rate in finding the ground truth rule, with 13% of these discoveries occurring during the abduction stage and 30% during the induction stage. In contrast, LLM agents exhibit a different pattern. They successfully identify the ground truth rule in approximately 30% of puzzles, with nearly all of these discoveries occurring during the abduction stage and only 3% achieved through interaction and induction. This highlights a significant limitation of current LLM agents, as they lack the ability to learn effectively from interactions. Consequently, the rule-learning patterns of LLM agents differ markedly from those of humans.

## A.2  TABLES

Table 4: Function operator puzzle distribution

| Puzzle ID | No. of Functions | No. of Unknown Paramters | No. of Elementary Functions |
|---|---|---|---|
| 1–5 | 1 | 1 | 1 |
| 6–10 | 2 | 2 | 2 |
| 11–15 | 2 | 2 | 3 |
| 16–20 | 3 | 3 | 4 |

Table 5: Escape room puzzle distribution, the number of paintings ranges from 3 to 13, increasing by one for each subsequent puzzle ID within each group. The first group (1–10) presents all paintings initially, whereas the second group (11–20) requires agent actions to reveal all paintings.

| Puzzle ID | Number of Paintings | Paintings Visibility |
|---|---|---|
| 1–10 | 3–13 | All paintings visible at start |
| 11–20 | 3–13 | Requires actions to view all paintings |

Table 6: Reactor puzzle distribution, five unique target strings need to be synthesized (ACB, CCADD, CADEA, ABCDEF, FEADE) are uniformly distributed across each rule category.

| Puzzle ID | Reactor Rule |
|-----------|--------------|
| 1–5 | Simple Concatenation |
| 6–10 | Reverse Concatenation |
| 11–15 | Middle Insertion |
| 16–20 | Prefix Replacement |

Table 7: Average Number of Repeated Actions Per Puzzle: Repeating actions is a common pattern among LLM agents during rule-learning tasks. Even sophisticated models like GPT-4o often exhibit reduced action duplication when exploring environments using the IDEA agent. The implementation of this agent has been shown to mitigate this tendency across all models. However, Gemma-7B frequently generates nonsensical actions that are not recognized as duplicates. Consequently, a duplication rate of 0.02 does not necessarily indicate that Gemma-7B effectively avoids repeating historical actions.

| Setup | Model | All Puzzles | Function Operator | Escape Room | Reactor Puzzles |
|-------|-------|-------------|-------------------|-------------|-----------------|
| Deduction Only | Gemma-7B | 8.28 | 3.75 | 11.49 | 9.59 |
| | Llama-8B | 2.39 | 3.4 | 1.2 | 2.56 |
| | Llama-70B | 2.38 | 2.91 | 0.44 | 3.8 |
| | GPT-3.5-Turbo | 6.18 | 8.79 | 5.26 | 4.5 |
| | GPT-4o | 1.75 | 1.55 | 0.34 | 3.35 |
| Baseline | Gemma-7B | 7.27 | 0.61 | 11.73 | 9.46 |
| | Llama-8B | 3.05 | 3.38 | 1.99 | 3.77 |
| | Llama-70B | 2.47 | 1.81 | 0.92 | 4.68 |
| | GPT-3.5-Turbo | 5.97 | 7.16 | 2.64 | 8.12 |
| | GPT-4o | 2.03 | 1.5 | 0.11 | 4.48 |
| IDEA | Gemma-7B | 2.74 | 6.33 | 1.88 | ~~0.02~~ |
| | Llama-8B | 1.83 | 2.29 | 1.64 | 1.56 |
| | Llama-70B | 1.01 | 0.45 | 0.39 | 2.18 |
| | GPT-3.5-Turbo | 4.49 | 4.14 | 2.4 | 6.93 |
| | GPT-4o | 1.12 | 0.52 | 0.15 | 2.68 |
| | Human | 0.76 | 0.46 | 1.6 | 0.22 |

## A.3 IDEA AGENT DETAIL

### A.3.1 ENVIRONMENT ENTITIES

- **Agent**: Represents the entity focused on rule-learning and problem-solving, comprising the following components:

  * **Goal** ($G$): The objective of the agent, articulated in natural language.

  * **Buffer Memories** ($\tilde{\mathbb{M}} := \{\tilde{m}_1, \tilde{m}_2, \ldots, \tilde{m}_n\}$): This temporary storage holds all newly generated information during the agent's exploration, including actions taken, outcomes of each action, and observations. This is where the most recent activities are initially recorded.

  * **Memories** ($\mathbb{M} := \{m_1, m_2, \ldots, m_n\}$): This permanent memory stores all significant observations and facts from the beginning of the task. When the agent forms new assumptions and plans, the contents of the Buffer Memories are evaluated; non-essential details like are discarded, while important facts and observations are transferred to the permanent Memories. This architecture ensures that each time the agent revises its hypotheses, it can clearly distinguish which observations occurred under the new assumptions and plan.

  * **Hypotheses** ($H$): The current hypotheses formulated by the agent to explain all the observations, are expressed in natural language.

  * **Plan** ($P$): The current strategy devised by the agent to either test the correctness of the existing hypotheses or to leverage these hypotheses to achieve the goal, also represented in natural language.

* **Action Space** ($\mathbb{A}$): A set of potential actions available to the agent, determined by its current hypotheses and plan. The Action Space is dynamic and can change in response to interactions with the environment. For example, after investigating a fridge, the agent gains the additional option to open the fridge and inspect its contents.

- **Objects** ($\mathbb{O}$): Represents all interactive entities within the environment that provide the agent with a means to interact with the world. A single object in this set is denoted as $\mathbf{O}$.

    * **Description** ($\mathbf{D_o}$): A concise description of the object, detailing its nature and potential uses, presented in natural language.

    * **Predefined interactive actions** ($\mathbf{O_A} := \{\tilde{\mathbf{a}}_1, \tilde{\mathbf{a}}_2, \ldots, \tilde{\mathbf{a}}_n\}$): A set of actions that are predefined for each object. Each action is described in natural language, explaining its purpose. Additionally, each action is associated with a coded function that processes the agent's inputs and produces an effect, potentially altering the environment based on these inputs.

### A.3.2 INTERACTIVE FUNCTIONS

- **Perceptual Action**:= $\hat{\mathbf{a}}(\mathbf{O})$: An action automatically added to the agent's action space for all objects within the same scope. Upon perceiving an object, the agent gains the ability to interact more detailedly with it, adding its interactive actions to the $\mathbf{S}$.

- **Interactive Action**:= $\tilde{\mathbf{a}}(\mathbf{D_o}, \mathbf{G}, \mathbf{H}, \mathbf{P}, \mathbf{I}, \tilde{\mathbb{M}}, \mathbb{M})$: A predefinec action that triggers a pre-coded effect based on the agent's input $\mathbf{I}$. For example, in using a reactor, the agent decides the materials and their order of addition, and the reactor processes these inputs based on pre-coded rules to synthesize new materials.

- **Abductive Action**:= $\bar{\mathbf{a}}(\mathbf{G}, \tilde{\mathbb{M}})$: An action based on initial observations, allowing the agent to formulate the first hypotheses and generate a new plan.

- **Inductive Action**:= $\dot{\mathbf{a}}(\mathbf{G}, \mathbf{H}, \mathbf{P}, \tilde{\mathbb{M}}, \mathbb{M})$: An action based on the current observations, goals, prior hypotheses, and previous plans, allowing the agent to refine hypotheses and generate new plans.

- **Deductive Action** := $\ddot{\mathbf{a}}(\mathbf{G}, \mathbf{H}, \mathbb{A}, \mathbb{M}, \tilde{\mathbb{M}})$: An action based on the current memories, hypothesis, and action space that generates a plan to either validate the current hypothesis or leverage it to solve problems.

- **Action select**:= $F_a(\mathbf{G}, \mathbf{H}, \mathbf{P}, \tilde{\mathbb{M}}, \mathbb{M}, \mathbb{A}) \to \mathbf{a}$: A function where the agent selects an action from the action space, considering all gathered information.

With the definitions and entities described above, we can formalize our interactive, rule-learning process. The sequence begins with the agent selecting an action from the available action space. The agent then decides on an input based on the selected action. Once the action is executed, the environment responds by providing feedback to the agent. The outcome of this action results in changes to $\tilde{\mathbb{M}}, \mathbb{M}, \mathbb{S}, \mathbf{H}, \mathbf{P}$ and $\mathbb{O}$, making the environment dynamic as the exploration process progresses. These changes reflect the agent's interactions and adaptations to the evolving conditions within the environment.

### A.3.3 PSEUDOCODE OF INTERACTIVE RULE LEARNING PROCEDURE

---

**Algorithm 2** Agent rule-learning procedure

---

1: **procedure** RULELEARNINGLOOP
2:     Initialize $\mathbb{O}, \mathbb{A}, \mathbf{G}$
3:     $\tilde{\mathbb{M}} \leftarrow$ Initial Memories
4:     $\mathbb{M} \leftarrow []$
5:     $\mathbf{H} \leftarrow \bar{\mathbf{a}}(\mathbf{G}, \tilde{\mathbb{M}}, \mathbb{A})$
6:     $\mathbf{P} \leftarrow \ddot{\mathbf{a}}(\mathbf{G}, \mathbf{H}, \mathbb{A}, \mathbb{M}, \tilde{\mathbb{M}})$
7:     $\tilde{\mathbb{M}}$.add("You established a new $\mathbf{H}$ and $\mathbf{P}$.")
8:     #step $\leftarrow 0$                                  ▷ Initialize step counter
9:     **while G** not achieved and step_count $\leq$ max_step **do**
10:         $\mathbf{a} \leftarrow F_a(\mathbf{G}, \mathbf{H}, \mathbf{P}, \tilde{\mathbb{M}}, \mathbb{M}, \mathbb{A})$         ▷ Select an action from the action space
11:         **if a** is a perceptual action **then**
12:             action_result $\leftarrow$ execute_perceptual_action($\mathbf{a}, \mathbf{O}$)
13:             $\mathbb{A} \leftarrow$ update_action_space(action_result)
14:             $\tilde{\mathbb{M}}$.add(action_result)                     ▷ Record result to buffer memory
15:         **else if a** is an interactive action **then**
16:             $\mathbf{I} \leftarrow$ decide_input($\mathbf{a}, \mathbf{D_o}, \mathbf{G}, \tilde{\mathbb{M}}, \mathbb{M}, \mathbf{H}, \mathbf{P}$)     ▷ Agent decide Input for this action
17:             action_result $\leftarrow$ execute_interactive_action($\mathbf{a}, \mathbf{I}$)
18:             $\mathbb{O} \leftarrow$ update_states(action_result)           ▷ update state of interactive objects
19:             $\mathbb{A} \leftarrow$ update_action_space(action_result)       ▷ Action may change action space
20:             #step $=$ #step $+ 1$          ▷ **Only interactive action increment step count**
21:             $\tilde{\mathbb{M}}$.add(action_result)                     ▷ Record result to buffer memory
22:         **else if a** is an inductive action **then**
23:             $\mathbf{H} \leftarrow \dot{\mathbf{a}}(\mathbf{G}, \mathbf{H}, \mathbf{P}, \tilde{\mathbb{M}}, \mathbb{M})$
24:             $\mathbf{P} \leftarrow \ddot{\mathbf{a}}(\mathbf{G}, \mathbf{H}, \mathbb{A}, \tilde{\mathbb{M}}, \mathbb{M})$
25:             $\mathbb{M}$.filter_add($\tilde{\mathbb{M}}$)       ▷ Drop non-observational log and add the rest to $\mathbb{M}$
26:             $\tilde{\mathbb{M}} \leftarrow []$                                  ▷ Empty buffer memory
27:             $\tilde{\mathbb{M}}$.add("You established a new $\mathbf{H}$ and $\mathbf{P}$.")
28:         **end if**
29:     **end while**
30: **end procedure**

---

## A.4 PROMPT EXAMPLE

### A.4.1 FUNCTION OPERATOR PUZZLES

**Action Select Prompt:**
**GOAl:** You are Kevin. You need to assign values to the functions displayed on the <Computer>, determine the values of 'a' and 'b'. Then, input these values into the <Code secured door> in alphabetical order to open it. You can test your hypothesis by entering values into the door. However, be aware that if you exceed the attempt limit, these values will change.
Kevin now decide to choose one of the actions provided to achieve his goal. Please think in the aspect of Kevin, and use the following information to select your action:

Following is the actions that Kevin did previously:
**You assign the value 1 to x of the function #1, and then the function outputs 6. (Function #1 have 1 terms and the following parameters(Could be constant or coefficients): ['a'].) ......**

After previous exploration, you have the following hypothesis and plan:
**Hypothesis and plan......**

Following is the x most recent things that Kevin have done under your current hypothesis:
**Most recent explorations guided by latest hypothesis and plan......**

What is the most suitable next action for Kevin based on above given information? Below are the available actions:
**1th action: Input code to the Code secured door and try opening it**
**2th action: Assign a value to the variable of Function #1 and see the output. Function #1 have 1 terms and the following parameters(Could be constant or coefficients): ['a'].**
**3th action: Assign a value to the variable of Function #2 and see the output. Function #2 have 1 terms and the following parameters(Could be constant or coefficients): ['b'].**
**4th action: Modify previous hypothesis and make a new plan: (Take this action when your current observations contradict your previous hypothesis for your current plan is fulfilled.)**

Above 4 provided actions are not yet performed by Kevin don't assume its outcome, please following the steps to generate your final answer. You MUST select one of the provided actions. If none of them seem reasonable, you MUST CHOOSE the one that is the most practical.
**Step1:** Review all the provided actions. Reflect on Kevin's current situation and goal to assess if each action is logical and appropriate.
**Step2:** Choose the most logical action. Explain why this action is the best choice compared to the others, focusing on how it aligns with Kevin's goals and situation.
**Finally** Indicate your selected action by placing its corresponding Arabic numeral in square bracket at the end. For example, if the third action is chosen, write [3]. Please do not use square bracket anywhere else other than final answer.
**Agent generated answer: ......**

**Abduction & Deduction Prompt**
**GOAL:** Same as left...
Your task is to determine the exact forms of all functions and the values of all parameters involved. First, focus on your observations to identify how many terms are in each function, the parameters within each, and any possible sub-functions involved in this puzzle. Then, hypothesize the actual forms of each function, including the values of constants and coefficients. Next, describe your plan for further verification, what value would you want to assign to which function, or do you want to input the password to the <Code secured door> to test your current result. Your response should include your current hypothesis and your planned actions.

Following is the actions that Kevin did previously:
**You assign the value 1 to x of the function #1, and then the function outputs 6. (Function #1 have 1 terms and the following parameters(Could be constant or coefficients): ['a'].) ......**

**Action Space**

**Agent generated answer: ......**

Figure 8: Prompt of Function Operator Puzzles, Action select and Deduction.

**Select Input Prompt**
**GOAl:** You are Kevin. You need to assign values to the functions displayed on the <Computer>, determine the values of 'a' and 'b'. Then, input these values into the <Code secured door> in alphabetical order to open it. You can test your hypothesis by entering values into the door. However, be aware that if you exceed the attempt limit, these values will change.
Kevin's current action is: Assign a value to the variable of Function #2 and see the output. Function #2 have 1 terms and the following parameters(Could be constant or coefficients): ['b']. And you have the following information to decide what is the value you want to assign to the variable in Function #2:

Following is the actions that Kevin did previously:
**You assign the value 1 to x of the function #1, and then the function outputs 6. (Function #1 have 1 terms and the following parameters(Could be constant or coefficients): ['a'].) ......**

After previous exploration, you have the following hypothesis and plan:
**Hypothesis and plan......**

Following is the x most recent things that Kevin have done under your current hypothesis:
**Most recent explorations guided by latest hypothesis and plan......**

Please follow the following steps to generate your final answer.
**Step 1:** Reflect on your recent experience and consider which value for the variable 'x' would be most helpful in decoding the parameters. State your reason of choosing this value. **Step 2:** Please enter the value you wish to assign to 'x', enclosed in square brackets, you can input 'pi', '0.5*pi', 'any_integer*pi', 'any_decimal*pi', any decimal, and any integer. For example, if you want to assign the value 12 to 'x', you should type ['12']. If you want to assign the value 0.5pi to 'x', you should type ['0.5*pi']. Please do not use fraction and "/" mark, please use decimal multiplication instead.

**Generated_answer:** "Step 1": "Assign x=1 to Function #2 to isolate 'b' since the output equals b/x, making the output equal to 'b'. This helps us find the exact value of 'b', simplifying our calculations for Function #1.",
  "Step 2": "[1]"

**Agent generated answer: ......**

**Induction & Deduction Prompt**
**GOAL:** Same as left...
Your task is to validate and modify your previous hypothesis, detailed here:
**Previous Hypothesis and plan......**

Determine if your current hypothesis is still valid. If it is, describe the next steps you plan to take towards your goal. If it is not, revise your hypothesis to accurately reflect all observations, both recent and prior. Finally, provide a plan for your next steps. Your response should include both your current hypothesis and your planned actions.

Following is the actions that Kevin did previously:
**You assign the value 1 to x of the function #1, and then the function outputs 6. (Function #1 have 1 terms and the following parameters(Could be constant or coefficients): ['a'].) ......**

**Action Space......**

**Agent generated answer: ......**

Figure 9: Prompt of Function Operator Puzzles, Interactive input and Induction.

**All Action History:**

**1,** You entered Puzzle room. Currently there are following items in this room: <Computer>,<Code secured door>

**2,** Kevin checked <Code secured door>, and found the result: This <Code secured door> is the only exit from this room. To unlock it and leave, you must enter the correct code. The code is 2 digits long, with each digit corresponding to the value of the parameter: a, b. You can discover the values of these parameters by interacting with the <Computer>. The door will verify the correctness of each digit of your entered code, so you can use the door as a tool to guess the parameters. However, if you fail 4 times, the parameter values will change.

**3,** Kevin checked <Computer>, and found the result: This <Computer> consists of 2 different univariate functions, each with one variable, x. Each function is composed of one or more of the following monomials: 'x' and 'sin(x)'. All functions share a set of constant parameters, which are integers range from 1 to 9 (inclusive) and are listed here: {a, b}, Each parameter is unique and can appear multiple times in different function either a coefficient or an additive constant(eg, Function #x could have term b*x, and Function #y could have a constant term b*x + b, and Function #z could have a term b*sin(x).) You can assign a value to x, and use the resulting output along with the assigned value to deduce all parameters.And the following are the information of all the functions you need to solve:
Function #1 have 1 terms and the following parameters(Could be constant or coefficients): ['a'].
Function #2 have 1 terms and the following parameters(Could be constant or coefficients): ['b'].

**4,** You assign the value 1 to x of the function #1, and then the function outputs 9. (Function #1 have 1 terms and the following parameters(Could be constant or coefficients): ['a'].)

**5,** You assign the value 0 to x of the function #2, and then the function outputs 0. (Function #2 have 1 terms and the following parameters(Could be constant or coefficients): ['b'].)

**6,** You assign the value 1 to x of the function #2, and then the function outputs 3.365883939231586. (Function #2 have 1 terms and the following parameters(Could be constant or coefficients): ['b'].)

**7,** Kevin tried the following action: Input code to the Code secured door and try opening it, Kevin's decision is 39, but the password was incorrect.Your first digit 3 is incorrect. Your second digit 9 is incorrect. (You can still try 3 times before the password change!)

**8,** You assign the value 0.5*pi to x of the function #2, and then the function outputs 4.0. (Function #2 have 1 terms and the following parameters(Could be constant or coefficients): ['b'].)

**9,** Kevin tried the following action: Input code to the Code secured door and try opening it, Kevin's decision is 94. The password is correct, Puzzle Finished!

**Hypothesis and Plan History:**

**Initial hypothesis and plan:**

**Hypothesis:** My current hypothesis is that Function #1 is a*x and Function #2 is b*sin(x).

**Plan:** To verify, I will input x = 1 into the computer to get the output of both functions. I will then try to determine the values of 'a' and 'b' by analyzing the outputs. If my hypothesis is correct, I will input the values of 'a' and 'b' in alphabetical order into the <Code secured door> to test my result.

**Induction**

**Second hypothesis and plan:**

**Hypothesis:** My current hypothesis holds: Function #1 is a*x and Function #2 is b*sin(x). From the outputs, I deduce that a = 9 (since 1*x = 9) and b = 3 (since 1*sin(1) ≈ 3.365883939231586).

**Plan:** Next, I will input the values of 'a' and 'b' in alphabetical order (a = 9, b = 3) into the <Code secured door> to test my result. If correct, the door will open. If not, I will reevaluate my hypothesis and adjust accordingly.

Figure 10: An example of Function Operator Puzzles is provided where actions marked in green are interactive actions, while the rest are perceptual actions through which the agent reads and perceives necessary environmental information.

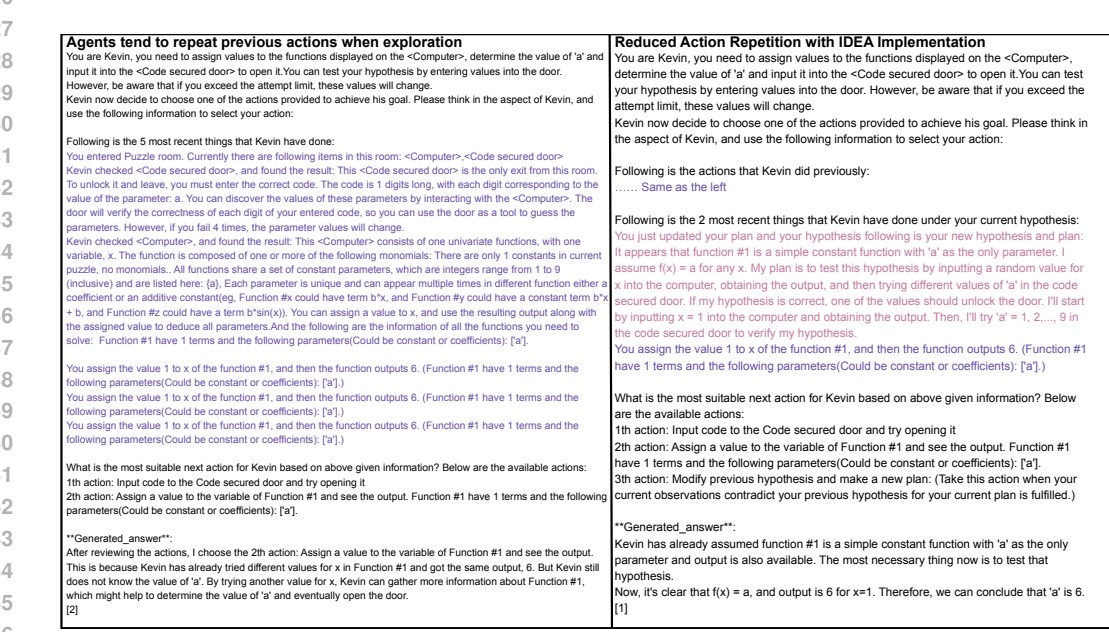

Figure 11: In the provided example, despite extensive exploration, the agent continues to assign multiple values to the function. In contrast, the IDEA agent hypothesizes that the function is simple, possessing only a single constant parameter. Consequently, this agent efficiently solves the puzzle by assigning just one value to the function.

## A.4.2 ESCAPE ROOM PUZZLES

**Action Select Prompt:**
**GOAL:** In this puzzle, set in an art gallery, Kevin must uncover the password for the <Code Secured Door> by discovering the relationships between the password and the paintings. And finally input the password into the <Code Secured door>. You can test your hypothesis by entering the password into the door. However, be aware that if you exceed the attempt limit, the password and hint will change.

Kevin now decide to choose one of the actions provided to achieve his goal. Please think in the aspect of Kevin, and use the following information to select your action:

Following is the actions that Kevin did previously:
**Kevin tried the following action: Input code to the Code Secured Door and try opening it, Kevin's decision is 421, but the password was incorrect.Kevin's first digit 4 is incorrect. Kevin's second digit 2 is correct. Kevin's third digit 1 is incorrect. ……**

After previous exploration, you have the following hypothesis and plan:
**Hypothesis and plan……**

Following is the 5 most recent things that Kevin have done under your current hypothesis:
**Most recent explorations guided by latest hypothesis and plan……**

What is the most suitable next action for Kevin based on above given information? Below are the available actions:

**1th action: Input code to the Code Secured Door and try opening it**
**2th action: Try opening the Code Secured Door with brute force**
**3th action: Modify previous hypothesis and make a new plan: (Take this action when your current observations contradict your previous hypothesis for your current plan is fulfilled.)**

Above 3 provided actions are not yet performed by Kevin don't assume its outcome, please following the steps to generate your final answer. You MUST select one of the provided actions. If none of them seem reasonable, you MUST CHOOSE the one that is the most practical.
**Step1:** Review all the provided actions. Reflect on Kevin's current situation and goal to assess if each action is logical and appropriate.
**Step2:** Choose the most logical action. Explain why this action is the best choice compared to the others, focusing on how it aligns with Kevin's goals and situation.
**Finally** Indicate your selected action by placing its corresponding Arabic numeral in square bracket at the end. For example, if the third action is chosen, write [3]. Please do not use square bracket anywhere else other than final answer.
**Agent generated answer: ……**

**Abduction & Deduction Prompt**
**GOAL:** In this puzzle, set in an art gallery, Kevin must uncover the password for the <Code Secured Door> by discovering the relationships between the password and the paintings. And finally input the password into the <Code Secured door>. Currently, you see from a note on the ground that says: You can test your hypothesis by entering the password into the door. However, be aware that if you exceed the attempt limit, the password and hint will change.
Your task is to formulate an hypothesis explaining how the password for the <Code Secured Door> relates to all the paintings in the gallery. Consider the observations provided and propose an initial hypothesis that accounts for your findings. Ensure your hypothesis is robust and consistent with all observations. Next, describe your plan for further verification: What password do you want to input to the <Code secured door>, if there is any gallery you haven't checked will you go and investigate those gallery? Your response should include your current hypothesis and your planned actions.

Following is the actions that Kevin did previously:
**Kevin tried the following action: Input code to the Code Secured Door and trying opening it, Kevin's decision is 421, but the password was incorrect.Kevin's first digit 4 is incorrect. Kevin's second digit 2 is correct. Kevin's third digit 1 is incorrect. ……**

**Action Space……**

**Agent generated answer: ……**

Figure 12: Prompt of Escape Room puzzles, Action select and Abduction.

**Select Input Prompt**
**GOAL:** In this puzzle, set in an art gallery, Kevin must uncover the password for the <Code Secured Door> by discovering the relationships between the password and the paintings. And finally input the password into the <Code Secured door>. You can test your hypothesis by entering the password into the door. However, be aware that if you exceed the attempt limit, the password and hint will change.
Kevin's current action is: Input code to the Code Secured Door and try opening it And you have the following information to decide what is the correct password:

Following is the actions that Kevin did previously:
Kevin tried the following action: Input code to the Code Secured Door and try opening it, Kevin's decision is 421, but the password was incorrect.Kevin's first digit 4 is incorrect. Kevin's second digit 2 is correct. Kevin's third digit 1 is incorrect. ……

After previous exploration, you have the following hypothesis and plan:
**Hypothesis and plan……**

Following is the x most recent things that Kevin have done under your current hypothesis:
**Most recent explorations guided by latest hypothesis and plan……**

Please follow the following steps to generate your final answer.
**Step1** reflect the recent experience, what do you think is the password to <Code Secured Door> is? Please only use information provided to do inference and give your reason. **Final Step** Please generate your final answer in a pair of square brackets. eg, if you think the final password is '999' you should output ['999'], if you think the output is '090' please output ['090'].

**Agent generated answer: ……**

**Induction & Deduction Prompt**
**GOAL:** Same as left…
Your task is to validate and modify your previous hypothesis, detailed here:
**Previous Hypothesis and plan……**

Determine if your current hypothesis is still valid. If it is, describe the next steps you plan to take towards your goal. If it is not, revise your hypothesis to accurately reflect all observations, both recent and prior. Finally, provide a plan for your next steps. Your response should include both your current hypothesis and your planned actions.

Following is the actions that Kevin did previously:
Kevin tried the following action: Input code to the Code Secured Door and try opening it, Kevin's decision is 421, but the password was incorrect.Kevin's first digit 4 is incorrect. Kevin's second digit 2 is correct. Kevin's third digit 1 is incorrect. ……

**Action Space……**

**Agent generated answer: ……**

Figure 13: Prompt of Escape Room puzzles, Interactive Input and Induction.

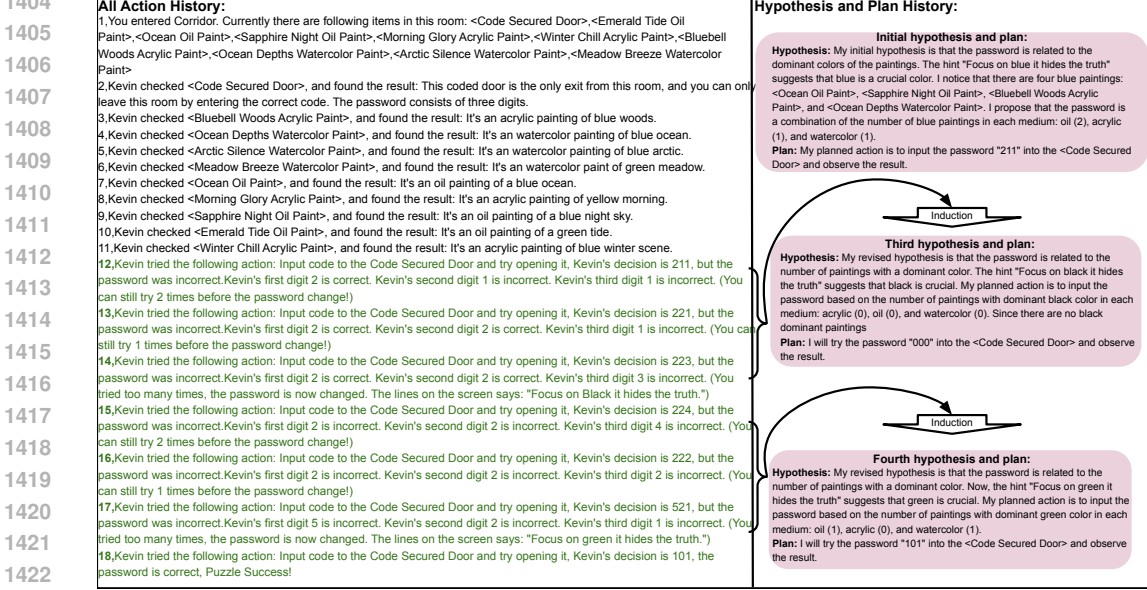

**All Action History:**

1,You entered Corridor. Currently there are following items in this room: <Code Secured Door>,<Emerald Tide Oil Paint>,<Ocean Oil Paint>,<Sapphire Night Oil Paint>,<Morning Glory Acrylic Paint>,<Winter Chill Acrylic Paint>,<Bluebell Woods Acrylic Paint>,<Ocean Depths Watercolor Paint>,<Arctic Silence Watercolor Paint>,<Meadow Breeze Watercolor Paint>

2,Kevin checked <Code Secured Door>, and found the result: This coded door is the only exit from this room, and you can only leave this room by entering the correct code. The password consists of three digits.

3,Kevin checked <Bluebell Woods Acrylic Paint>, and found the result: It's an acrylic painting of blue woods.

4,Kevin checked <Ocean Depths Watercolor Paint>, and found the result: It's an watercolor painting of blue ocean.

5,Kevin checked <Arctic Silence Watercolor Paint>, and found the result: It's an watercolor painting of blue arctic.

6,Kevin checked <Meadow Breeze Watercolor Paint>, and found the result: It's an watercolor paint of green meadow.

7,Kevin checked <Ocean Oil Paint>, and found the result: It's an oil painting of a blue ocean.

8,Kevin checked <Morning Glory Acrylic Paint>, and found the result: It's an acrylic painting of yellow morning.

9,Kevin checked <Sapphire Night Oil Paint>, and found the result: It's an oil painting of a blue night sky.

10,Kevin checked <Emerald Tide Oil Paint>, and found the result: It's an oil painting of a green tide.

11,Kevin checked <Winter Chill Acrylic Paint>, and found the result: It's an acrylic painting of blue winter scene.

**12,**Kevin tried the following action: Input code to the Code Secured Door and try opening it, Kevin's decision is 211, but the password was incorrect.Kevin's first digit 2 is correct. Kevin's second digit 1 is incorrect. Kevin's third digit 1 is incorrect. (You can still try 2 times before the password change!)

**13,**Kevin tried the following action: Input code to the Code Secured Door and try opening it, Kevin's decision is 221, but the password was incorrect.Kevin's first digit 2 is correct. Kevin's second digit 2 is correct. Kevin's third digit 1 is incorrect. (You can still try 1 times before the password change!)

**14,**Kevin tried the following action: Input code to the Code Secured Door and try opening it, Kevin's decision is 223, but the password was incorrect.Kevin's first digit 2 is correct. Kevin's second digit 2 is correct. Kevin's third digit 3 is incorrect. (You tried too many times, the password is now changed. The lines on the screen says: "Focus on Black it hides the truth.")

**15,**Kevin tried the following action: Input code to the Code Secured Door and try opening it, Kevin's decision is 224, but the password was incorrect.Kevin's first digit 2 is incorrect. Kevin's second digit 2 is incorrect. Kevin's third digit 4 is incorrect. (You can still try 2 times before the password change!)

**16,**Kevin tried the following action: Input code to the Code Secured Door and try opening it, Kevin's decision is 222, but the password was incorrect.Kevin's first digit 2 is incorrect. Kevin's second digit 2 is incorrect. Kevin's third digit 2 is incorrect. (You can still try 1 times before the password change!)

**17,**Kevin tried the following action: Input code to the Code Secured Door and try opening it, Kevin's decision is 521, but the password was incorrect.Kevin's first digit 5 is incorrect. Kevin's second digit 2 is incorrect. Kevin's third digit 1 is incorrect. (You tried too many times, the password is now changed. The lines on the screen says: "Focus on green it hides the truth.")

**18,**Kevin tried the following action: Input code to the Code Secured Door and try opening it, Kevin's decision is 101, the password is correct, Puzzle Success!

**Hypothesis and Plan History:**

**Initial hypothesis and plan:**

**Hypothesis:** My initial hypothesis is that the password is related to the dominant colors of the paintings. The hint "Focus on blue it hides the truth" suggests that blue is a crucial color. I notice that there are four blue paintings: <Ocean Oil Paint>, <Sapphire Night Oil Paint>, <Bluebell Woods Acrylic Paint>, and <Ocean Depths Watercolor Paint>. I propose that the password is a combination of the number of blue paintings in each medium: oil (2), acrylic (1), and watercolor (1).

**Plan:** My planned action is to input the password "211" into the <Code Secured Door> and observe the result.

*Induction*

**Third hypothesis and plan:**

**Hypothesis:** My revised hypothesis is that the password is related to the number of paintings with a dominant color. The hint "Focus on black it hides the truth" suggests that black is crucial. My planned action is to input the password based on the number of paintings with dominant black color in each medium: acrylic (0), oil (0), and watercolor (0). Since there are no black dominant paintings

**Plan:** I will try the password "000" into the <Code Secured Door> and observe the result.

*Induction*

**Fourth hypothesis and plan:**

**Hypothesis:** My revised hypothesis is that the password is related to the number of paintings with a dominant color. Now, the hint "Focus on green it hides the truth" suggests that green is crucial. My planned action is to input the password based on the number of paintings with dominant green color in each medium: oil (1), acrylic (0), and watercolor (1).

**Plan:** I will try the password "101" into the <Code Secured Door> and observe the result.

Figure 14: An example of Escape Room is provided where actions marked in green are interactive actions. The remaining actions, which are perceptual, allow the agent to read and gather necessary environmental information. In the given example, although the agent successfully guesses the correct rule behind the observations, it fails to adhere to its plan and assumptions. When inputting the password, the attempts do not align with the planned strategy, and it also makes repeated attempts (repeated 101 twice) that yield no useful results.

### A.4.3 REACTOR PUZZLES

**Action Select Prompt:**

**GOAL:** You are Kevin. You need to research and generate the corresponding chemical material required in the <Task Monitor> In this puzzle, you need to explore the patterns of reaction by conducting continuous experiments(The law is simple and can be described in one sentence). Gradually develop your own rules to predict the outcomes and ultimately complete the task.You know from an incomplete list of reaction equations that: XY+Z = ZXY.

Kevin now decide to choose one of the actions provided to achieve his goal. Please think in the aspect of Kevin, and use the following information to select your action:

Following is the actions that Kevin did previously:

**By turning on the reactor B and C turned into CB after the reaction. And you put the products into your storage for later use. .......**

After previous exploration, you have the following hypothesis and plan:
**Hypothesis and plan……**

Following is the 5 most recent things that Kevin have done under your current hypothesis:
**Most recent explorations guided by latest hypothesis and plan……**

You currently have the following items in your storage: <A>, , <C> ……(All synthesized material)

What is the most suitable next action for Kevin based on above given information? Below are the available actions:

**1th action: Choose material you want to synthesize, and turn on the Reactor.**
**2th action: Modify previous hypothesis and make a new plan: (Take this action when your current observations contradict your previous hypothesis for your current plan is fulfilled.)**

Above 2 provided actions are not yet performed by Kevin don't assume its outcome, please following the steps to generate your final answer. You MUST select one of the provided actions. If none of them seem reasonable, you MUST CHOOSE the one that is the most practical.

**Step1:** Review all the provided actions. Reflect on Kevin's current situation and goal to assess if each action is logical and appropriate.

**Step2:** Choose the most logical action. Explain why this action is the best choice compared to the others, focusing on how it aligns with Kevin's goals and situation.

**Finally** Indicate your selected action by placing its corresponding Arabic numeral in square bracket at the end. For example, if the third action is chosen, write [3]. Please do not use square bracket anywhere else other than final answer.

**Agent generated answer: ……**

**Abduction & Deduction Prompt**

**GOAL:** You are Kevin. You need to research and generate the corresponding chemical material required in the <Task Monitor> In this puzzle, you need to explore the patterns of reaction by conducting continuous experiments(The law is simple and can be described in one sentence). Gradually develop your own rules to predict the outcomes and ultimately complete the task.You know from an incomplete list of reaction equations that: XY+Z = ZXY.

Your task is to formulate an hypothesis based on the reactions you observe. Please use the given observations to propose an initial rule that explains all reactions observed. Ensure your hypothesis is robust and consistent with these reactions. Next, describe your plan for further verification: which two materials from the following list will you use to test your hypothesis? Available materials: You currently have the following items in your storage: <A>, , <C>. Your response should include your current hypothesis and your planned actions.

Following is the actions that Kevin did previously:
**By turning on the reactor B and C turned into CB after the reaction. And you put the products into your storage for later use. .......**

**Action Space……**

**Agent generated answer: ……**

Figure 15: Prompt of Reactor Puzzles, Action select and Abduction

**Select Input Prompt**

**GOAL:** You are Kevin. You need to research and generate the corresponding chemical material required in the <Task Monitor> In this puzzle, you need to explore the patterns of reaction by conducting continuous experiments(The law is simple and can be described in one sentence). Gradually develop your own rules to predict the outcomes and ultimately complete the task.You know from an incomplete list of reaction equations that: XY+Z = ZXY.

Kevin's current action is: Choose material you want to synthesize, and turn on the Reactor. And you have the following information to decide what material you put into the reactor:

Following is the actions that Kevin did previously:
By turning on the reactor B and C turned into CB after the reaction. And you put the products into your storage for later use. …….

After previous exploration, you have the following hypothesis and plan:
**Hypothesis and plan……**

Following is the x most recent things that Kevin have done under your current hypothesis:
**Most recent explorations guided by latest hypothesis and plan……**

Please follow the steps below to decide which materials you should put into the reactor.

**Step 1:**Given all the material in the storage you can use and synthetics you require to create:You currently have the following items in your storage: ……
Decide which (one or two) material you want to put into the reactor this time you can select any material from your storage, you need to clear specify the reaction you excepted and state the formula.
**Step 2:** Please copy the name of the selected material and paste the name into a pair of parentheses, and separate two different material with comma. The name should be exactly as provided, enclosed in parentheses, for example, if you want to put a unit of X and a unit of Y into the reactor and make an reaction, please answer (X, Y), if you want to see what comes out the reactor with material <XY> and <Z> you should answer(XY, Z). You can only choose the material that listed in your storage. Please do not forget the parentheses!

**Agent generated answer: ……**

**Induction & Deduction Prompt**

**GOAL:** Same as left…
Your task is to validate and modify your previous hypothesis, detailed here:
**Previous Hypothesis and plan……**

Determine if your current hypothesis is still valid. If it is, describe the next steps you plan to take towards your goal. If it is not, revise your hypothesis to accurately reflect all observations, both recent and prior. Finally, provide a plan for your next steps. Your response should include both your current hypothesis and your planned actions.

Following is the actions that Kevin did previously:
By turning on the reactor B and C turned into CB after the reaction. And you put the products into your storage for later use. …….

**Action Space……**

**Agent generated answer: ……**

Figure 16: Prompt of Reactor puzzles, Interactive input and Induction.

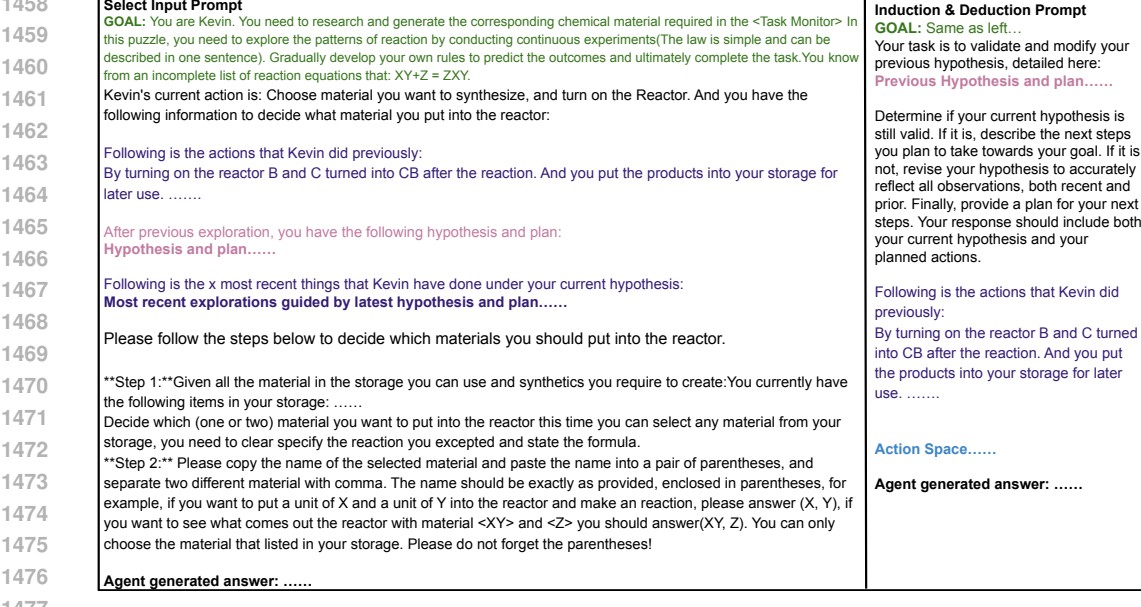

**All Action History:**

**1,**You entered Chemical laboratory. Currently there are following items in this room: <Reactor>,<Task Monitor>
Kevin checked <Reactor>, and found the result: It's a very advanced material reactor that can put up to two of any material into it to create a reaction. The reactor provided unlimited one-letter raw material and other materials you've already synthesized.

**2,By turning on the reactor A and B turned into BA after the reaction. And you put the products into your storage for later use.**
**3,By turning on the reactor BA and C turned into CBA after the reaction. And you put the products into your storage for later use.**
**4,By turning on the reactor C and BA turned into BAC after the reaction. And you put the products into your storage for later use.**
**5,By turning on the reactor BAC and A turned into ABAC after the reaction. And you put the products into your storage for later use.**
**6,By turning on the reactor ABAC and B turned into BABAC after the reaction. And you put the products into your storage for later use.**
**7,By turning on the reactor BAC and A turned into ABAC after the reaction. And you put the products into your storage for later use.**
**8,By turning on the reactor BAC and A turned into ABAC after the reaction. And you put the products into your storage for later use.**
**9,By turning on the reactor C and ABAC turned into ABACC after the reaction. And you put the products into your storage for later use.**
**10,By turning on the reactor A and C turned into CA after the reaction. And you put the products into your storage for later use.**
**11,By turning on the reactor CA and B turned into BCA after the reaction. And you put the products into your storage for later use.**

**Hypothesis and Plan History:**

**Initial hypothesis and plan:**
**Hypothesis:** Assuming that the reaction follows a concatenation rule, where the reactants are combined in the order they are added, I propose an initial hypothesis: "When combining two materials, the resulting material is created by concatenating the reactants in the order they are added.
**Plan:** To verify this, I will test my hypothesis by reacting <A> and  to form <AB>, and then reacting <AB> with <C> to form <ABC>. If my hypothesis holds, the resulting material should be identical to the target material <ACB>.

Induction

**Second hypothesis and plan:**
**Hypothesis:** The reaction follows a concatenation rule, but with a twist - the second reactant is appended to the beginning of the first reactant.
**Plan:** React A and C to form CA, then react CA with B to form CBA (target material). If successful, this will validate my revised hypothesis. If not, I'll reassess and adjust my approach. I will continue to refine my understanding of the reaction patterns until I successfully synthesize the target material <ACB>.

Figure 17: An example of Reactor Puzzles is provided where actions marked in green are interactive actions. The remaining actions are perceptual, allowing the agent to read and gather necessary environmental information. In the given example, the agent eventually realizes the flaws in its initial hypothesis and generates a correct one. However, the agent reaches the step limit before it can implement the solution, failing.

### A.4.4 HULLICINATION EXAMPLES

**Agent's new observation contradict to current hypothesis**

You are Kevin. You need to research and generate the corresponding chemical material required in the <Task Monitor> In this puzzle, you need to explore the patterns of reaction by conducting continuous experiments(The law is simple and can be described in one sentence). Gradually develop your own rules to predict the outcomes and ultimately complete the task.You know from an incomplete list of reaction equations that: XY+Z = ZXY.

Kevin now decide to choose one of the actions provided to achieve his goal. Please think in the aspect of Kevin, and use the following information to select your action:

Following is the actions that Kevin did previously:
You entered Chemical laboratory. Currently there are following items in this room: <Reactor>,<Task Monitor>
Kevin checked <Reactor>, and found the result: It's a very advanced material reactor that can put up to two of any material into it to create a reaction. The reactor provided unlimited one-letter raw material and other materials you've already synthesized.
Kevin checked <Task Monitor>, and found the result: It's a quest display that shows you the current quest you need to complete, which currently reads, "Please synthesise a unit of <CADEA> materials and put it into the slot under the monitor."
After previous exploration, you have the following hypothesis and plan:
"hypothesis": "When combining two materials, the second material is added at the beginning of the first material's sequence.",
"plan": "To verify this hypothesis, I will put material <A> first and then material <C> into the reactor.", please act based on your hypothesis and plan.

Following is the 1 most recent things that Kevin have done under your current hypothesis:
By turning on the reactor A and C turned into CA after the reaction. And you put the products into your storage for later use.
You currently have the following items in your storage: <A>, , <C>, <D>, <E>, <CA>

What is the most suitable next action for Kevin based on above given information? Below are the available actions:
1th action: Choose material you want to synthesize, and turn on the Reactor.
2th action: Modify previous hypothesis and make a new plan: (Take this action when your current observations contradict your previous hypothesiss or your current plan is fulfilled.)
Above 2 provided actions are not yet performed by Kevin don't assume its outcome, please following the steps to generate your final answer. You MUST select one of the provided actions. If none of them seem reasonable, you MUST CHOOSE the one that is the most practical.
**Step1:** Review all the provided actions. Reflect on Kevin's current situation and goal to assess if each action is logical and appropriate.
**Step2:** Choose the most logical action. Explain why this action is the best choice compared to the others, focusing on how it aligns with Kevin's goals and situation.
**Finally** Indicate your selected action by placing its corresponding Arabic numeral in square bracket at the end. For example, if the third action is chosen, write [3]. Please do not use square bracket anywhere else other than final answer.

**Agent fail to identify the contradiction**

**Generated_answer**
"Step1": "Reviewing the two provided actions, the first action involves synthesizing a new material with the Reactor, while the second action involves modifying the hypothesis or plan. Kevin's observation so far aligns with his hypothesis.",
"Step2": "Given that the hypothesis has been verified (CA from A and C), Kevin should continue with the synthesis to achieve the goal of CADEA. Utilizing available materials and continuing with described pattern CA lead to CA -> CAA and onwards towards CADEA.",
"Selected_Action": "[1]"

The agent has already verified its hypothesis by synthesizing CA from A + C; however, in the next step, the agent tends to synthesize CAA, which is not part of the final target product. Despite knowing the rule, the agent fails to leverage this rule effectively towards achieving the final goal.

Figure 18: Example of GPT-4o agent failing to leverage rule towards the result.

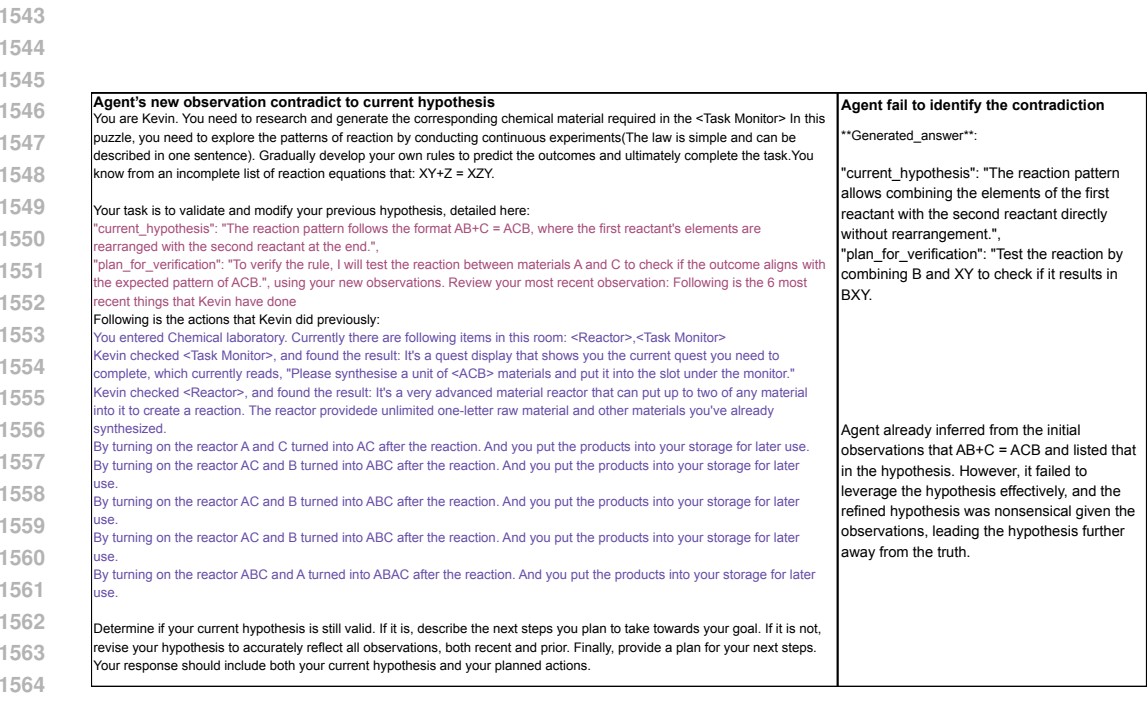

Figure 19: Example of hallucination by GPT-4o when refining previous hypothesis.

| Agent's new observation contradict to current hypothesis | Agent fail to identify the contradiction |
|---|---|
| You are Kevin. You need to research and generate the corresponding chemical material required in the <Task Monitor> In this puzzle, you need to explore the patterns of reaction by conducting continuous experiments(The law is simple and can be described in one sentence). Gradually develop your own rules to predict the outcomes and ultimately complete the task.You know from an incomplete list of reaction equations that: XY+Z = ZXY.
Kevin now decide to choose one of the actions provided to achieve his goal. Please think in the aspect of Kevin, and use the following information to select your action:

Following is the actions that Kevin did previously:
You entered Chemical laboratory. Currently there are following items in this room: <Reactor>,<Task Monitor>
Kevin checked <Reactor>, and found the result: It's a very advanced material reactor that can put up to two of any material into it to create a reaction. The reactor provided unlimited one-letter raw material and other materials you've already synthesized.
Kevin checked <Task Monitor>, and found the result: It's a quest display that shows you the current quest you need to complete, which currently reads, "Please synthesise a unit of <CCADD> materials and put it into the slot under the monitor."

After previous exploration, you have the following hypothesis and plan:
Initial Hypothesis: Materials combine in a consistent, alphabetical order, resulting in the longest possible string of unique letters.
To test this hypothesis, I will combine <A> and  to see if AB is produced, and then combine <C> and <D> to see if CD is produced. If my hypothesis holds, these reactions will occur, and I can further refine my rule., please act based on your hypothesis and plan.

Following is the 1 most recent things that Kevin have done under your current hypothesis:
By turning on the reactor A and B turned into BA after the reaction. And you put the products into your storage for later use.

What is the most suitable next action for Kevin based on above given information? Below are the available actions:
1th action: Choose material you want to synthesize, and turn on the Reactor.
2th action: Modify previous hypothesis and make a new plan: (Take this action when your current observations contradict your previous hypothesis or your current plan is fulfilled.)

Above 2 provided actions are not yet performed by Kevin don't assume its outcome, please following the steps to generate your final answer. You MUST select one of the provided actions. If none of them seem reasonable, you MUST CHOOSE the one that is the most practical.
**Step1:** Review all the provided actions. Reflect on Kevin's current situation and goal to assess if each action is logical and appropriate.
**Step2:** Choose the most logical action. Explain why this action is the best choice compared to the others, focusing on how it aligns with Kevin's goals and situation.
**Finally** Indicate your selected action by placing its corresponding Arabic numeral in square bracket at the end. For example, if the third action is chosen, write [3]. Please do not use square bracket anywhere else other than final answer. | **Generated_answer**:

Upon reviewing the provided actions, I find that action 1 is the most logical choice. Since Kevin's goal is to synthesize CCADD and he has already tested his initial hypothesis by combining A and B, the next step would be to continue testing his hypothesis by combining C and D. This aligns with his plan to refine his rule and ultimately achieve his goal. Kevin can now proceed to combine C and D in the reactor. [1]



The agent hypothesized that the materials combine in a consistent, alphabetical order; however, its most recent experiment shows the result A+B = BA, which contradicts this hypothesis. Noticing the contradiction, the agent should choose the second action to refine its hypothesis rather than continue to synthesize the materials under the incorrect hypothesis. |

Figure 20: Example of Llama-3 70B agent failing to detect a contradiction in experimental results.

| Context and prompt for agent inputting password in Room Escape puzzle | Agent's answer |
|---|---|
| Based on the following recent experience of Kevin:
In this puzzle, set in an art gallery, Kevin must uncover the password for the <Code Secured Door> by discovering the relationships between the password and the paintings. And finally input the password into the <Code Secured door>. Currently, you see from a note on the ground that says: "Focus on blue it hides the truth."You can test your hypothesis by entering the password into the door. However, be aware that if you exceed the attempt limit, the password and hint will change.
Kevin's current action is: Input code to the Code Secured Door and try opening it And you have the following information to decide what is the correct password:

Following is the 13 most recent things that Kevin have done:
You entered Corridor. Currently there are following items in this room: <Code Secured Door>,<Flower Oil Paint>,<Lemon Grove Oil Paint>,<Coastal Serenity Oil Paint>,<Sapphire Night Oil Paint>,<Jungle Mist Acrylic Paint>,<Winter Chill Acrylic Paint>,<Arctic Silence Watercolor Paint>
Kevin checked <Code Secured Door>, and found the result: This coded door is the only exit from this room, and you can only leave this room by entering the correct code. The password consists of three digits.
Kevin checked <Coastal Serenity Oil Paint>, and found the result: It's an oil painting of a blue coast.
Kevin checked <Arctic Silence Watercolor Paint>, and found the result: It's an watercolor painting of blue arctic.
Kevin checked <Lemon Grove Oil Paint>, and found the result: It's an oil painting of a yellow lemon grove.
Kevin checked <Flower Oil Paint>, and found the result: It's an oil painting of a yellow flower.
Kevin checked <Sapphire Night Oil Paint>, and found the result: It's an oil painting of a blue night sky.
Kevin checked <Jungle Mist Acrylic Paint>, and found the result: It's an acrylic painting of green jungle.
Kevin checked <Winter Chill Acrylic Paint>, and found the result: It's an acrylic painting of blue winter scene.
Kevin tried the following action: Input code to the Code Secured Door and try opening it, Kevin's decision is 123, but the password was incorrect.Kevin's first digit 1 is incorrect. Kevin's second digit 2 is incorrect. Kevin's third digit 3 is incorrect.
You can still try 2 times before the password change!
Kevin tried the following action: Input code to the Code Secured Door and try opening it, Kevin's decision is 258, but the password was incorrect.Kevin's first digit 2 is correct. Kevin's second digit 5 is incorrect. Kevin's third digit 8 is incorrect.
You can still try 1 times before the password change!
Please follow the following steps to generate your final answer.
**Step1** reflect on the recent experience, what do you think is the password to <Code Secured Door> is? Please only use information provided to do inference and give your reason. **Final Step** Please generate your final answer in a pair of square brackets. eg, if you think the final password is '123' you should output [123], if you think the output is '324' please output ['324']. | Based on the recent experience, I think the password to <Code Secured Door> is 289. My reason is that the hint "Focus on blue it hides the truth" suggests that the correct password is related to the blue-colored paintings. The correct first digit 2 is likely from the blue <Coastal Serenity Oil Paint> or <Winter Chill Acrylic Paint>. The correct second digit 8 is likely from the blue <Sapphire Night Oil Paint>. The correct third digit 9 is likely from the blue <Arctic Silence Watercolor Paint>.

[289] |

Figure 21: Example of hallucination by Llama-3 70B when generating a password.

## A.5 PUZZLE EXAMPLES

**Function Operator Puzzle No. 2 (3 functions, 2 unknown parameters, 2 elementary functions)**

**Goal:** You are Kevin. You need to assign values to the functions displayed on the <Computer>, determine the values of 'a' and 'b'. Then, input these values into the <Code secured door> in alphabetical order to open it.You can test your hypothesis by entering values into the door. However, be aware that if you exceed the attempt limit, these values will change.

**Initial Memories:**

1, Kevin checked <Code secured door>, and found the result: This <Code secured door> is the only exit from this room. To unlock it and leave, you must enter the correct code. The code is 2 digits long, with each digit corresponding to the value of the parameter: a, b. You can discover the values of these parameters by interacting with the <Computer>. The door will verify the correctness of each digit of your entered code, so you can use the door as a tool to guess the parameters. However, if you fail 4 times, the parameter values will change.

2, Kevin checked <Computer>, and found the result: This <Computer> consists of 2 different univariate functions, each with one variable, x. Each function is composed of one or more of the following monomials: "x^2" (square of x) and 'sin(x)'. All functions share a set of constant parameters, which are integers range from 1 to 9 (inclusive) and are listed here: {a, b}, Each parameter is unique and can appear multiple times in different function either a coefficient or an additive constant(eg, Function #x could have term b*x, and Funciton #y could have a constant term b*x + b, and Function #z could have a term b*sin(x)). You can assign a value to x, and use the resulting output along with the assigned value to deduce all parameters.And the following are the information of all the functions you need to solve:

Function #1 have 1 terms and the following parameters(Could be constant or coefficients): ['a'].

Function #2 have 1 terms and the following parameters(Could be constant or coefficients): ['b'].

**Provided Interactive actions:**

1th action: Input code to the Code secured door and try opening it

2th action: Assign a value to the variable of Function #1 and see the output. Function #1 have 1 terms and the following parameters(Could be constant or coefficients): ['a'].

3th action: Assign a value to the variable of Function #2 and see the output. Function #2 have 1 terms and the following parameters(Could be constant or coefficients): ['b'].

Figure 22: Function operator puzzle No. 2.

**Function Operator Puzzle No. 17 (3 functions, 3 unknown parameters, 4 elementary functions)**

**Goal:** You are Kevin. You need to assign values to the functions displayed on the <Computer>, determine the values of 'a', 'b', 'c'. Then, input these values into the <Code secured door> in alphabetical order to open it.You can test your hypothesis by entering values into the door. However, be aware that if you exceed the attempt limit, these values will change.

**Initial Memories:**

1, Kevin checked <Code secured door>, and found the result: This <Code secured door> is the only exit from this room. To unlock it and leave, you must enter the correct code. The code is 3 digits long, with each digit corresponding to the value of the parameter: a, b, c. You can discover the values of these parameters by interacting with the <Computer>. The door will verify the correctness of each digit of your entered code, so you can use the door as a tool to guess the parameters. However, if you fail 4 times, the parameter values will change.

2, Kevin checked <Computer>, and found the result: This <Computer> consists of 3 different univariate functions, each with one variable, x. Each function is composed of one or more of the following monomials: '|x|' (absolute value of x),'x',"x^2" (square of x) and 'sin(x)'. All functions share a set of constant parameters, which are integers range from 1 to 9 (inclusive) and are listed here: {a, b, c}, Each parameter is unique and can appear multiple times in different function either a coefficient or an additive constant(eg, Function #x could have term b*x, and Function #y could have a constant term b*x + b, and Function #z could have a term b*sin(x)). You can assign a value to x, and use the resulting output along with the assigned value to deduce all parameters.And the following are the information of all the functions you need to solve:

Function #1 have 2 terms and the following parameters(Could be constant or coefficients): ['a', 'b'].

Function #2 have 2 terms and the following parameters(Could be constant or coefficients): ['a', 'c'].

Function #3 have 1 terms and the following parameters(Could be constant or coefficients): ['c'].

**Provided Interactive actions:**

1th action: Input code to the Code secured door and try opening it

2th action: Assign a value to the variable of Function #2 and see the output. Function #2 have 2 terms and the following parameters(Could be constant or coefficients): ['a', 'c'].

3th action: Assign a value to the variable of Function #1 and see the output. Function #1 have 2 terms and the following parameters(Could be constant or coefficients): ['a', 'b'].

4th action: Assign a value to the variable of Function #3 and see the output. Function #3 have 1 terms and the following parameters(Could be constant or coefficients): ['c'].

Figure 23: Function operator puzzle No. 17.

**Escape Room Puzzle No. 3 (6 Paintings, All Paintings visible)**

**Goal:** In this puzzle, set in an art gallery, Kevin must uncover the password for the <Code Secured Door> by discovering the relationships between the password and the paintings. And finally input the password into the <Code Secured door>. You can test your hypothesis by entering the password into the door. However, be aware that if you exceed the attempt limit, the password and hint will change.

**Initial Memories:**

1, You entered Corridor. Currently there are following items in this room: <Code Secured Door>,<Emerald Tide Oil Paint>,<Sapphire Night Oil Paint>,<Bluebell Woods Acrylic Paint>,<Morning Glory Acrylic Paint>,<Arctic Silence Watercolor Paint>,<River Reflections Watercolor Paint>

2, Kevin checked <Code Secured Door>, and found the result: This coded door is the only exit from this room, and you can only leave this room by entering the correct code. The password consists of three digits.

3, Kevin checked <Bluebell Woods Acrylic Paint>, and found the result: It's an acrylic painting of blue woods.

4, Kevin checked <Arctic Silence Watercolor Paint>, and found the result: It's an watercolor painting of blue arctic.

5, Kevin checked <River Reflections Watercolor Paint>, and found the result: It's an watercolor painting of blue river.

6, Kevin checked <Morning Glory Acrylic Paint>, and found the result: It's an acrylic painting of yellow morning.

7, Kevin checked <Sapphire Night Oil Paint>, and found the result: It's an oil painting of a blue night sky.

8, Kevin checked <Emerald Tide Oil Paint>, and found the result: It's an oil painting of a green tide.

9, Currently, you see from a note on the ground that says: "Focus on blue it hides the truth."

**Provided Interactive actions:**

1th action: Input code to the Code Secured Door and try opening it

2th action: Try opening the Code Secured Door with brute force

Figure 24: Escape room puzzle No. 3

**Escape Room Puzzle No. 13 (6 Paintings, Need to actively explore the gallery to reveal all paintings)**

**Goal:** In this puzzle, set in an art gallery, Kevin must uncover the password for the <Code Secured Door> by discovering the relationships between the password and the paintings. And finally input the password into the <Code Secured door>. You can test your hypothesis by entering the password into the door. However, be aware that if you exceed the attempt limit, the password and hint will change.

**Initial Memories:**

1, You entered Oil Painting Gallery. Currently there are following items in this room: <Emerald Tide Oil Paint>,<Sapphire Night Oil Paint>,<Code Secured Door>,<Watercolour Gallery Entrance>,<Acrylic Painting Gallery Entrance>

2, Kevin checked <Code Secured Door>, and found the result: This coded door is the only exit from this room, and you can only leave this room by entering the correct code. The password consists of three digits.

3, Kevin checked <Watercolour Gallery Entrance>, and found the result: It's an automatic door with a poster next to it that says "Watercolour Gallery".

4, Kevin checked <Acrylic Painting Gallery Entrance>, and found the result: It's an automatic door with a poster next to it that says "Acrylic Painting Gallery."

5, Kevin checked <Sapphire Night Oil Paint>, and found the result: It's an oil painting of a blue night sky.

6, Kevin checked <Emerald Tide Oil Paint>, and found the result: It's an oil painting of a green tide.

7, Currently, you see from a note on the ground that says: "Focus on blue it hides the truth."

**Provided Interactive actions:**

1th action: Input code to the Code Secured Door and try opening it

2th action: Try opening the Code Secured Door with brute force

3th action: Pass through the Watercolour Gallery Entrance and reaches the Watercolour Gallery.

4th action: Pass through the Acrylic Painting Gallery Entrance and reaches the Acrylic Painting Gallery.

Figure 25: Escape room puzzle No. 13

**Reactor Puzzle No. 8 (Reverse concatenation rule, Target material: "CADEA")**

**Goal:** You are Kevin. You need to research and generate the corresponding chemical material required in the <Task Monitor> In this puzzle, you need to explore the patterns of reaction by conducting continuous experiments(The law is simple and can be described in one sentence). Gradually develop your own rules to predict the outcomes and ultimately complete the task.

**Initial Memories:**

1, Kevin checked <Reactor>, and found the result: It's a very advanced material reactor that can put up to two of any material into it to create a reaction. The reactor provided unlimited one-letter raw material and other materials you've already synthesized.

2, Kevin checked <Task Monitor>, and found the result: It's a quest display that shows you the current quest you need to complete, which currently reads, "Please synthesise a unit of <CADEA> materials and put it into the slot under the monitor."

3, You currently have the following items in your storage: <A>, , <C>, <D>, <E>.

4, You know from an incomplete list of reaction equations that: XY+Z = ZXY.

**Provided Interactive actions:**

1th action: Choose material you want to synthesize, and turn on the Reactor.

---

**Reactor Puzzle No. 14 (Middle insertion rule, Target material: "ABCDEF")**

**Goal:** You are Kevin. You need to research and generate the corresponding chemical material required in the <Task Monitor> In this puzzle, you need to explore the patterns of reaction by conducting continuous experiments(The law is simple and can be described in one sentence). Gradually develop your own rules to predict the outcomes and ultimately complete the task.

**Initial Memories:**

1, Kevin checked <Reactor>, and found the result: It's a very advanced material reactor that can put up to two of any material into it to create a reaction. The reactor provided unlimited one-letter raw material and other materials you've already synthesized.

2, Kevin checked <Task Monitor>, and found the result: It's a quest display that shows you the current quest you need to complete, which currently reads, "Please synthesise a unit of <ABCDEF> materials and put it into the slot under the monitor."

3, You currently have the following items in your storage: <A>, , <C>, <D>, <E>, <F>

4, You know from an incomplete list of reaction equations that: XY+Z = XZY.

**Provided Interactive actions:**

1th action: Choose material you want to synthesize, and turn on the Reactor.

Figure 26: Reactor puzzle No. 8 and No. 14

