# OpenReview forum: "IDEA: Enhancing the Rule Learning Ability of Large Language Model Agent through Induction, Deduction, and Abduction"
_ICLR.cc/2025/Conference — ICLR 2025 Conference Withdrawn Submission_

### Official Review · Reviewer_L1re · 2024-10-26

**Soundness:** 3
**Presentation:** 3
**Contribution:** 2
**Rating:** 6
**Confidence:** 4

**Summary:**

The paper introduces a benchmark for assessing the rule learning abilities of LLMs in an interactive environment. The paper also proposes a reasoning framework that does abduction, induction, and deduction in order to do well in such interactive environments. The framework outperforms the baseline, although has not reached human performance in the tasks.

**Strengths:**

The benchmark introduced in this paper is reasonably timely, pushing towards open language description of problems that require interactive problem solving. The proposed framework of abduction, followed by induction and deduction is reasonable and showed improved performance over the baseline. Some interesting observations are made in the human studies, e.g. that humans do not do well initially and appear not to do much abduction. This can lead to interesting followup work. Overall, I think that this is useful work that is competently done and can help advance the area.

**Weaknesses:**

Overall, the benchmark is okay for exploring interactively solving problems described in natural language. But it is not highly compelling. It is not as targetted towards human priors like the ARC challenge which targets objectness, goal-directness, number and counting, and basic geometry and topology. It is also not clearly useful like the Behaviour-1K benchmark. The size of the benchmark is also not large, so it is useful for exploring the problem with more benchmarks required for verifying the capabilities tested.

**Questions:**

It would be helpful if the capabilities the benchmark is testing can be clearly stated, e.g. see the priors stated in the ARC challenge https://arcprize.org/arc. More insights on the human problem solving strategies used by the human participants would be useful since they appear not to do well in abduction but are more successful at the end.

---

> ### Author Response · Authors · 2024-11-19
> **Respond to Weakness and Questions**
>
> Thank you for your comments and suggestions. We would like to address your concerns as follows:
>
> ### **Weakness**
> - Our benchmark, RULEARN, differs fundamentally from the ARC challenge by requiring agents to actively gather observations through strategic interaction with the environments to learn rules and solve tasks. In the ARC challenge, agents are provided with several input-output pairs and are tasked with predicting the output for a new input without needing to design experiments or gather additional observations.
>     - For example, the tasks in ARC are like:
>         - Fig1 -> Output Fig1
>         - Fig2 -> Output Fig2
>         - Fig3 -> Output Fig3
>         - Fig4 -> ?
>         - The agent then generates the target output.
>     - If ARC were a defined puzzle in RULEARN, the task would shift as follows:
>         - Given a Target Output
>         - **Agent itself draws** Fig1 -> Output Fig1 ≠ Target Output
>             - Guessing the transformation rule between Figure 1 and Output Figure 1, the agent formulates Hypothesis 1 and plans to draw new figures to verify this hypothesis.
>         - **Agent itself draws** Fig2 -> Output Fig2 ≠ Target Output
>             - The agent refines Hypothesis 1 based on the new observation, forming Hypothesis 2 and adjusting the plan.
>         - … (Steps continue)
>         - Agent finds out the transformation rule between input figures and output figures.
>         - Agent draws Figure n → Output Figure n = Target Output (Success)
>     - In this scenario, agents must design their own experiments to uncover the underlying transformation rules. This process mirrors human cognitive strategies where we formulate hypotheses and test them through experimentation. Unlike ARC, where agents receive all necessary observation pairs upfront, RULEARN requires agents to actively seek out information and iteratively refine hypothesized rules, making it a more challenging and realistic test of rule-learning abilities.
> - Required Prior Knowledge:
>     - Our benchmark also necessitates that agents have basic prior knowledge relevant to each puzzle type:
>         - In **Function Operator puzzles**, agents need to know basic mathematics.
>         - To solve **Escape Room puzzles**, agents must have a grasp of numbers, counting and grouping, as they need to count paintings with different attributes to deduce the password.
>         - In **Reactor puzzles**, agents need to understand ordering and recognize how different letters are combined.
> - The Behaviour-1K benchmark focuses on robotic tasks and does not involve rule learning or strategic experimentation. Therefore, it serves a different purpose and should not be directly compared to RULEARN, which is centered on cognitive problem-solving and hypothesis-driven learning.
> - We will add discussion about these works in related works in revision.
>
> ### **Question**
> - As defined in the provided link, “The intelligence of a system is measured by its skill-acquisition efficiency over a range of tasks, with respect to priors, experience, and generalization difficulty.”
> - Our work surpasses ARC with this definition in the following way:
>         - In ARC, the agent is provided with all necessary observations and directly answers the question. We can only judge the intelligence of the agent based on whether the answer is right or wrong.
>         - In RULEARN, agents proactively gather new observations to learn rules and solve the task. We can evaluate not only how efficient (step count) an agent to successfully complete the task but also examine how many observations an agent typically needs to formulate a more accurate hypothesis. Additionally, we can observe the specific rules that the agent learns. By utilizing tasks defined in RULEARN, we can measure intelligence with a more detailed metric compared to ARC.
> - Furthermore, RULEARN is more extendable than ARC. While the information in ARC is presented using a colored grid, RULEARN utilizes natural language and features interactive objects that mimic real-life situations in detail. This allows us to define more complex puzzles that test the agent's specific prior knowledge more rigorously.

---

> ### Comment · Reviewer_L1re · 2024-11-24
>
> Thanks for the response. I maintain my view that this is nice work although not yet highly compelling, and lean towards accept.

---

### Official Review · Reviewer_oa2T · 2024-11-02

**Soundness:** 3
**Presentation:** 3
**Contribution:** 3
**Rating:** 6
**Confidence:** 4

**Summary:**

This paper proposes a benchmark (RULEARN) and a framework (IDEA) to evaluate the rule-learning capabilities of LLM agents in interactive settings. The proposed framework, IDEA, integrates induction, deduction, and abduction, where the LLM agent forms initial hypotheses, devises plans to validate these hypotheses, and refines previous hypotheses. The IDEA agent is evaluated on RULEARN across three domains: the Function Operator, the Escape Room, and the Reactor. Experiments show that IDEA agents significantly outperform baselines but still lag behind human participants.

**Strengths:**

- The paper provides a benchmark to evaluate various reasoning capabilities of LLMs in interactive settings. The proposed tasks, although mainly focused on puzzles, are interesting and serve the evaluation purpose. The results provide insights into LLMs' rule-learning capabilities in real-world scenarios.
- The experiments are comprehensive, covering a wide range of puzzle sets and five LLMs. Human experiments are also helpful.
- The paper includes various quantitative and qualitative analyses of the experimental results. Comparisons between human and model performance provide useful insights.
- The paper is clearly written and easy to follow.

**Weaknesses:**

- The naming of the Oracle-rule agent feels somewhat misleading to me. IMO, it is not really an oracle, as the LLM is not provided with the ground truth rule. Instead, the agent is simply solving an easier task with additional information about the rule. Therefore, this might not serve as a fair or useful upper bound, as the task is actually different. Based on L311-312, “Even if the agent could successfully learn the correct rule, applying the learned rule to solve the puzzle is non-trivial,” I assume the authors meant that the “meta-rule” is provided, but the agent still needs to determine the exact instantiation. I think this should be clarified further to avoid confusion.
- It would be interesting if the authors could include a breakdown of performance based on puzzle difficulty for different models. Difficulty could be measured using the statistics defined in Appendix A2, e.g., for the Function Operator, it could be the number of functions, etc. This could help readers understand how the performance of different models changes when varying difficulty levels.
- There are many studies evaluating LMs’ performance on text-based games, e.g. [1] and many follow-up works. In most of these settings, the LM agent needs to determine the goal and solve the game. What’s the difference between this paper and that line of work? Is the key difference that the rule is more fine-grained, or that the IDEA agent is explicitly evaluated on induction, abduction, and deduction? It would be beneficial if the authors could further discuss and clarify this.

[1] Marc-Alexandre Côté, Ákos Kádár, Xingdi Yuan, Ben Kybartas, Tavian Barnes, Emery Fine, James Moore, Ruo Yu Tao, Matthew Hausknecht, Layla El Asri, Mahmoud Adada, Wendy Tay, Adam Trischler. TextWorld: A Learning Environment for Text-based Games.

**Questions:**

- I’m curious about the connection between this paper and active learning. The interactive action defined in IDEA is similar to adding new examples. Some studies related to this line of work, e.g. [2], are not discussed in the paper. It would be helpful to discuss these connections and differences.

[2] Wasu Top Piriyakulkij, Cassidy Langenfeld, Tuan Anh Le, Kevin Ellis. Doing Experiments and Revising Rules with Natural Language and Probabilistic Reasoning.

---

> ### Author Response · Authors · 2024-11-19
> **Respond to weakness (1)**
>
> Thank you for your comments and suggestions. We would like to address your concerns as follows:
> ### **1, The naming of the Oracle-rule agent feels misleading.**
> - It is true that summarizing the "Ground truth" in natural language does not strictly make the agent an oracle. We initially thought the term "oracle" was straightforward, but we will consider a new name for this setting in the updated version, perhaps "Rule-Hinted Agent" or another suitable term.
> ### **2, It would be interesting if the authors could include a breakdown of performance based on puzzle difficulty for different models.**
> - To save space, we list the breakdown performance of GPT-4o here, since this breakdown is not very straightforward with table, we will add the visualization of each puzzle in our revision.
>     - Specifically, In function operator puzzles:
>         - Difficulty1 = 1 function, 1 parameter, and 1 elementary function
>         - Difficulty2 = 2 functions, 2 parameters, and 2 elementary functions
>         - Difficulty3 = 2 functions, 2 parameters, and 3 elementary functions
>         - Difficulty4 = 3 functions, 3 parameters, and 4 elementary functions
>     - In Escape room puzzles:
>         - Difficulty1 = 3~8 paints, All paints visible at the start
>         - Difficulty2 = 9~13 paints, All paints visible at the start
>         - Difficulty3 = 3~8 paints, Need to go to other rooms to investigate other paints
>         - Difficulty4 = 9~13 paints, Need to go to other rooms to investigate other paints
>     - In Reactor puzzles: (Reaction rule is not the same)
>         - Difficulty1 = Simple Concatenation
>         - Difficulty2 = Reverse Concatenation
>         - Difficulty3 = Middle insertion
>         - Difficulty4 = Perfix Replacement
> - Function_operator_puzzles, (element is the success puzzle count, total 25 puzzles in each setting)
>
>         - Setting   | Difficulty 1 | Difficulty 2 | Difficulty 3 | Difficulty 4
>         - Oracle    | 25              | 5                | 12             | 3
>         - Baseline  | 25              | 2                | 0              | 0
>         - IDEA      | 25              | 8                | 6              | 1
>
> - Escape_room_puzzles
>
>         - Setting   | Difficulty 1 | Difficulty 2 | Difficulty 3 | Difficulty 4
>         - Oracle    | 21              | 12               | 24             | 24
>         - Baseline  | 6               | 4                | 6              | 5
>         - IDEA      | 12              | 11               | 7              | 12
>
> - Reactor_puzzles
>
>         - Setting   | Difficulty 1 | Difficulty 2 | Difficulty 3 | Difficulty 4
>         - Oracle    | 24              | 4                | 4              | 1
>         - Baseline  | 18              | 1                | 2              | 0
>         - IDEA      | 23              | 1                | 0              | 0
> - As demonstrated above, most puzzles are solved primarily at Difficulty 1. Even advanced models like GPT-4o can only solve a small portion of the puzzles at higher difficulty levels. Other models exhibit a similar pattern.

---

> ### Author Response · Authors · 2024-11-19
> **Respond to weakness (2)**
>
> ### **3, What’s the difference between this paper and other text-based work?**
> - **Difference of the task and benchmark**: Our project was inspired by TextWorld, as I have long been a fan, and aimed to create a text-based environment that mimics the real world. However, I discovered that TextWorld does not support fine-grained interactions; the agent is only provided with basic actions like "take," "open," and "go to." With such a limited action space, it is challenging to devise tasks that mimic real-life rule discovery. To address this, we designed a new framework that allows easy editing and supports fine-grained interactions similar to those encountered in real life. This framework underpins RULEARN, which we discuss in the common questions section. In environments like TextWorld, which are constrained by coarse-grained action spaces, agents can only be tested with everyday tasks, such as placing an item into a specific location. These tasks are simple and straightforward, often not requiring significant reasoning—a human could perform these actions from muscle memory without much thought. Specifically, in other text-based environments, there is no "rule" to learn; all the information provided is clear and straightforward, focusing solely on using the provided information efficiently to solve tasks. However, in RULEARN, tasks are inherently challenging due to the presence of an unknown rule. To succeed, the agent must identify patterns from gathered observations, use the patterns to iteratively refine the hypothesized rules, and use the rule to solve the task. This process demands intensive reasoning and carefully planned trial-and-error attempts.
> - **Difference between the methods**: The methods used to solve text-based games in TextWorld and RULEARN differ significantly due to the nature of the tasks. Before the rise of LLMs, reinforcement learning (RL) was a commonly used approach for solving text-based games. However, we demonstrate that RL-based methods are not applicable to our RULEARN benchmark in the question answering part. With the advent of LLMs, many benchmarks became saturated with high-performing models, as these benchmarks often involve straightforward tasks. In contrast, our baseline shows that even with advanced prompting methods, LLMs still struggle in RULEARN. This is because they tend to rely on shallow reasoning and reflection, failing to think deeply or identify common patterns across observations. Our IDEA framework addresses this limitation by guiding LLM agents to adopt a human-like rule-learning loop, incorporating Abduction, Deduction, and Induction:
>     - **Abduction**: Generates an intial hypothesis to explain all initial observations.(Automatically called one time at the beginning to form an initial hypothesis)
>     - *(Input: initial observations → Output: a new hypothesis)*
>     - **Deduction**: Generates a plan to verify this hypothesis. (Automatically called when hypothesis is generated or refined)
>     - *(Input: hypothesis, goal → Output: plan)*
>     - **Induction**: Refines the previous hypothesis with new observations. (The agent proactively selects this action from the action space when it detects inconsistencies between the observations and the current hypothesis.)
>     - *(Input: old hypothesis, new observations, old observations → Output: refined hypothesis)*
> -  This approach enables agents to strategically explore the environment through experimentation and develop deeper reasoning to solve complex tasks.

---

> ### Author Response · Authors · 2024-11-19
> **Answering the questions**
>
> ### **Q1 I’m curious about the connection between this paper and active learning.**
> - In real-life scenarios, a hypothesis is sometimes just a vague idea and can be expressed in various ways. For instance, in the Reactor puzzle, upon observing the outcome "A + B -> AB," an agent could formulate semantically different hypotheses such as "The reaction rule simply combines the letters in the order provided," or "The reaction rule combines two letters in alphabetical order." Each hypothesis accurately explains the observation but implies a completely different underlying rule.
> - In fields like active learning or reinforcement learning, each step typically involves optimizing a clear mathematical scalar to select an action that maximizes rewards. However, when two semantically distinct rules explain the same observation, there is no straightforward mathematical scalar to determine which subsequent observation would optimize an objective. Although we might define an objective that aligns with these two specific hypotheses, the most remarkable aspect of human reasoning is our creativity, which allows us to explain observations in numerous ways.
> - For instance, the observation “A + B -> AB” can also be interpreted based on the number of closed spaces, since 'A' has one closed triangle and 'B' has two closed semicircles. This hypothesis would predict “AB + C -> CAB.” Holding different hypotheses in our minds leads us to anticipate different observations that confirm our theories. This necessitates the use of varied objective functions tailored to each hypothesis, with their definitions potentially varying significantly.
> - In real life, it is impossible to define a scalar metric to mathematically evaluate how effective the next observation will be, even if the rule is already well-understood. This is because we can still interpret a well-learned rule from different perspectives across various domains. In our RULEARN benchmark, although we know the ground truth, we cannot predict how an agent will interpret the observations. This uncertainty means that traditional methods of active learning and reinforcement learning are not directly applicable in our benchmark.

---

> > ### Comment · Reviewer_oa2T · 2024-11-23
> > **Response to the Authors**
> >
> > Thank you to the authors for the response! I encourage the authors to include some of the discussions here in the paper's next revision. I will maintain my score as accept.

---

### Official Review · Reviewer_gg78 · 2024-11-03

**Soundness:** 2
**Presentation:** 3
**Contribution:** 2
**Rating:** 3
**Confidence:** 4

**Summary:**

The paper proposed an interactive reasoning benchmark RULEARN, which focuses on evaluating LLM agents' rule-learning and planning abilities in an interactive environment. The benchmark contains three types of puzzles "function operator", where the agent needs to determine the parameters and element functions of a set of functions by trying different input values and observe output values; "escape room", where the agent needs to gather information in a textual environment and guess a code; "reactor" where the agent needs to figure out what a string operator do and come up with a plan to reach a specific string using the operator.

The paper also proposed a rule-learning agent, IDEA, which can perform abduction, induction, and deduction iteratively to explore an interactive environment, learn rules, and achieve goals.

**Strengths:**

- The paper proposed a benchmark to assess LLM agents' rule-learning ability in an interactive environment, which is different from existing reasoning benchmarks where there are no interactive information-seeking actions required.
- The paper proposed an LLM agent, IDEA, which achieves higher performance than the baseline reasoning method in the proposed benchmark.
- The presentation of the paper is clear and easy to follow.

**Weaknesses:**

While the idea of evaluating the reasoning ability of LLM agents in an interactive environment where information-seeking is necessary is interesting, there are several key drawbacks of the paper:
1. The size of the proposed benchmark is very small, with only twenty puzzles per problem. This will make the evaluation very noisy and also eliminate the possibility of any training on these tasks, which significantly limits the application of the benchmark.
2. The proposed IDEA agent needs prompt tuning for each specific problem, which limits its generalizability. Considering the proposed benchmark has limited test cases, the reviewer is worried that the good performance of the IDEA agent is a result of prompt tuning overfitting to the test problems.

**Questions:**

1. How is the Baseline agent implemented, how is the IDEA agent different from the baseline agent? It would be good to give a prompt example that highlights the key differences between the proposed IDEA agent and the baseline agent.
2. Why use the Wilson confidence interval instead of simply the success rate as the evaluation metric, since it's less intuitive to understand the performance of different methods?

---

> ### Author Response · Authors · 2024-11-19
> **Respond to weakness**
>
> Thank you for your comments and suggestions. We would like to address your concerns as follows:
> ### **1, Regarding the limited size of our benchmark**
> - See general response.
> ### **2, Need prompt tuning**
> - We initially conducted prompt tuning to help the agent generate formatted responses. During our early experiments, we discovered that such tuning did not significantly affect the results, so we ceased prompt tuning for the IDEA agent at an early stage of our study. Below, we provide the results recently rerun with the IDEA agent and GPT-4o using a tuned IDEA prompt (Shown in our paper) and our old untuned prompt with a temperature of 0.
> - To save space we split our prompt into a different post following.
> - New Result for IDEA agent (Since openai updated gpt-4o and their api since our original experiment in the paper we rerun our result with gpt-4o and the result shows as follows.)
>     - Oracle:
>         - All:(55.77\%, 78.69\%)
>         - Function operator:(58.4\%, 91.93\%)
>         - Escape room:(83.89\%, 100.0\%)
>         - Reactor puzzle:(11.19\%, 46.87\%)
>     - Baseline:
>         - All:(22.73\%, 45.94\%)
>         - Function operator:(29.93\%, 70.07\%)
>         - Escape room:(14.55\%, 51.9\%)
>         - Reactor puzzle:(8.07\%, 41.6\%)
>     - IDEA (Tuned version):
>         - All:(33.09\%, 57.51\%)
>         - Function operator:(43.29\%, 81.88\%)
>         - Escape room:(25.82\%, 65.79\%)
>         - Reactor puzzle:(8.07\%, 41.6\%)
>     - IDEA (Untuned version):
>         - All:(37.74\%, 62.26\%)
>         - Function operator:(58.4\%, 91.93\%)
>         - Escape room:(25.82\%, 65.79\%)
>         - Reactor puzzle:(11.19\%, 46.87\%)
> - As shown above, the results are robust to different prompts, because the prompts we provided only constrained the agent to generate formatted sentences and ensured no details were missed in any specific puzzle, without teaching them how to reason based on each puzzle type. If we were to further tune the agent on how to reason within the prompt according to puzzle type, we might expect better results. However, we did not pursue this to maintain a fair comparison.

---

> ### Author Response · Authors · 2024-11-19
> **Answering the questions (1)**
>
> ### **Q1: How is the Baseline agent implemented, how is the IDEA agent different from the baseline agent?**
> - Our baseline agent relies on conventional reasoning and reflective thinking to navigate puzzles. This approach mirrors methodologies like ReACT and Reflexion, where the agent's actions are guided by a sequential reasoning process, and each decision is informed by previous experiences and available information within the environment. For instance, here's how our baseline agent approaches one of the puzzles (Generated text of agent):
>     - …... previous steps
>     - **1, Step 1**: Reflecting on Kevin's recent experience, the note says "Focus on blue it hides the truth." Kevin has checked two blue paintings: <Midnight Dance Oil Paint> and <River Reflections Watercolor Paint>. The first painting is of a blue night sky with the moon, and the second is of a blue river. The password consists of three digits. Given the emphasis on blue and the fact that there are two blue paintings, it is likely that the digits are derived from these paintings. Since there are two blue paintings, the digits could be the number of blue paintings (2) and the number of words in the titles of these paintings (Midnight Dance = 2 words, River Reflections = 2 words). Thus, the password could be 222.**Final Step**: Generate the final answer in a pair of square brackets.[222] (Reflect/Reason on previous information agent decide the password is 222)
>     - …… few more steps
>     - **n, Step1:** Review all the provided actions. Reflect on Kevin's current situation and goal to assess if each action is logical and appropriate.**Step2:** Choose the most logical action. Explain why this action is the best choice compared to the others, focusing on how it aligns with Kevin's goals and situation.Kevin knows the first digit is 1 and the third digit is 1. The hint "Focus on blue it hides the truth" suggests the second digit is related to blue paintings. The blue paintings are "Midnight Dance" and "River Reflections," both of which are associated with the number 0 (as 2 and 3 were incorrect).**Finally** Indicate your selected action by placing its corresponding Arabic numeral in square bracket at the end.The most logical action is to input the code 101.[1] (Agent failed several times, reflect the attemptation experience, agent get closer to the answer and get a closer guess 101)
> - As shown, our baseline can be viewed as employing ReACT and Reflexion methodologies to solve puzzles sequentially. Each step requires the agent to reflect on all observations and previously taken actions before deciding on the next step.
> - The IDEA agent inherits all the prompts, including those for reasoning and reflecting at each step. The primary difference between IDEA and the baseline is that the IDEA agent introduce the explicit rule-learning loop of Abduction, Deduction, and Induction, with the following new actions added to the action space:
>     - **Abduction**: Generates a hypothesis to explain all initial observations.(Automatically called one time at the beginning to form an initial hypothesis)
>     - *(Input: initial observations → Output: a new hypothesis)*
>     - **Deduction**: Generates a plan to verify this hypothesis. (Automatically called when hypothesis is generated or refined)
>     - *(Input: hypothesis, goal → Output: plan)*
>     - **Induction**: Refines the previous hypothesis with new observations. (The agent proactively selects this action from the action space when it detects inconsistencies between the observations and the current hypothesis.)
>     - *(Input: old hypothesis, new observations, old observations → Output: refined hypothesis)*
> - Compared to the baseline agent (which could also be labeled as ReACT), the IDEA agent has these additional actions to form the hypothesis and create a new plan when inconsistencies are detected between observations and the hypothesis. When this action is selected, the agent reviews all observations and previous hypotheses, refines the hypothesis to ensure it aligns with all observations, drafts a new plan to verify the hypothesis, and proceeds with actions guided by this updated hypothesis and plan.
> - Both the baseline agent and IDEA agent use the same prompt structure, which can be reviewed in appendix A.4. The IDEA agent receives additional, hypothesis-related prompts during the abduction,  induction, and deduction phases, and the prompt is listed in the following post
>
> *[1] Shunyu Yao et al. ReAct: Synergizing Reasoning and Acting in Language Models.*
>
> *[2] Noah Shinn et al. Reflexion: Language Agents with Verbal Reinforcement Learning*

---

> ### Author Response · Authors · 2024-11-19
> **Answering the questions (2)**
>
> ### **Q2, Why use the Wilson confidence interval?**
> - The use of confidence intervals is crucial in distinguishing the performance gap between different models under various settings. For example, if we only use the average success rate, the results might appear as follows:
>    -  Baseline:
>         - All: 34.335%
>     - IDEA:
>         - All: 50%
> -  These numbers alone do not provide sufficient information to distinguish true performance since the 50% could be associated with high variance, and the 34.335% could be associated with low variance. However, including confidence intervals offers a more detailed view:
>     - Baseline:
>         - All:(22.73\%, 45.94\%)
>     - IDEA:
>         - All:(37.74\%, 62.26\%)
> - With confidence intervals, we can discern that:
>     - The IDEA exhibits slightly higher variance in its performance compared to the Baseline.
>     - We are 95% confident that the success rate of IDEA falls above 37.74%, which is significantly better than the average score of the Baseline.
>     - We are 95% confident that the success rate of the Baseline falls below 45.94%, which is lower than the average performance of IDEA.
>     - With more calculation, we are able to say we are 96.55% statistically confident that the IDEA outperforms the Baseline.
> - By introducing confidence intervals, we can more clearly understand each model's performance in different settings by considering the variation in performance.
> - In the future revision, we will use mean ± interval to replace our current expression to make it clearer.

---

> ### Author Response · Authors · 2024-11-19
> **Prompt for tuned and untuned IDEA**
>
> ### **Tuned abduction (used in our paper):**
> - Prompt for Function operator puzzle:
>     - Abduction: f'''{GOAL} Your task is to determine the forms of each functions and what value you can assign to x to reveal the values of the parameters. First, focus on your observations to identify how many terms are in each function, the parameters within each, and any possible sub-functions involved in this puzzle. Next, describe your plan for further verification, what value would you want to assign to which function, or do you want to input the password to the <Code secured door> to test your current result. Your response should include your current hypothesis and your planned actions. You can choose from the following actions:\n{action_space_str}\nPlease keep your analysis, hypothesis, and planned actions as concise and precise as possible.'''
> - Prompt for Escape room puzzles:
>     - Abduction: f'''{GOAL} Your task is to determine the forms of each functions and what value you can assign to x to reveal the values of the parameters. First, focus on your observations to identify how many terms are in each function, the parameters within each, and any possible sub-functions involved in this puzzle. Next, describe your plan for further verification, what value would you want to assign to which function, or do you want to input the password to the <Code secured door> to test your current result. Your response should include your current hypothesis and your planned actions. You can choose from the following actions:\n{action_space_str}\nPlease keep your analysis, hypothesis, and planned actions as concise and precise as possible.'''
> - Prompt for Reactor puzzles:
>     - Abduction: f'''{GOAL} Your task is to formulate an assumption based on the reactions you observe. Please use the given observations to propose an initial rule that explains all reactions observed. Ensure your assumption is robust and consistent with these reactions. Next, describe your plan for further verification: which two materials from the following list will you use to test your assumption? Available materials: {storage_information}. Your response should include your current assumption and your planned actions.'''
>
> ### **Untuned abduction (Initial prompt at the beginning of our study):**
> - Global abduction: f'''{GOAL} Your task is to develop general, clearly falsifiable, explanatory rules that explains your observations, a process known as abduction. Please consider the given observations and propose an initial hypothesis that explains them, make sure your hypothesis is robust and align with all your observations. Your response should include your current hypothesis and your planned actions.\nPlease keep your analysis, hypothesis, and planned actions as concise and precise as possible.'''
> ### **We use the same induction prompt at the beginning of our experiment across all puzzles:**
> - Global Induction: f'''{GOAL} Your task is to validate and modify your previous hypothesis, detailed here: {Previous_hypothesis}, using your new observations. Review your most recent observation: {buffer_memory_str}, to determine if your current hypothesis is still valid. If it is, describe the next steps you plan to take towards your goal. If it is not, revise your hypothesis to accurately reflect all observations, both recent and prior. Finally, provide a plan for your next steps. Your response should include both your current hypothesis and your planned actions. You can choose from the following actions:\n{action_space_str}\nPlease keep your analysis, hypothesis, and planned actions as concise and precise as possible.'''
> - The experiment results for other LLMs show similar trends. We can see that prompt tuning does not bring improvements. We will add these prompt details and results in our revision.

---

> > ### Comment · Reviewer_gg78 · 2024-11-26
> > **Thank you for your response**
> >
> > The response resolved some of my concerns about the prompt-tuning overfitting issue as the authors have provided the details of the tuned and untuned prompts.
> >
> > I understand that the scale of the dataset is restricted by the financial costs of conducting the experiments. However, I remain pessimistic about the applicability and usefulness of the proposed benchmark due to its limited scale and toyish domain (function operator, escape room, and string combining rules). I would suggest the author improve the paper by 1) increasing the scale of the benchmark so that evaluation on it is more robust, and 2) finding more realistic domains for rule learning so that methods developed on it are more useful and convincing.
> >
> > For the above reasons, I will keep my original score.

---

### Official Review · Reviewer_fgiR · 2024-11-03

**Soundness:** 2
**Presentation:** 3
**Contribution:** 2
**Rating:** 3
**Confidence:** 4

**Summary:**

This paper constructs a benchmark, named RULEARN, to evaluate the rule-learning abilities of LLM agents in interactive settings and proposes a reasoning framework, named IDEA, which integrates the process of induction, deduction, and abduction. The authors evaluate five representative LLMs.
RULEARN contains three types of tasks: The Function Operator, The Escape Room, and The Reactor.
These tasks are all about reasoning some hidden rules.
IDEA first generate a hypothesis and then generates a plan based on the hypothesis, interacts with the environment to valid the hypothesis and then refine the initial hypothesis based on observations.

**Strengths:**

1. The authors construct an interactive environment containing three types of task, each has 20 puzzles.
2. The authors propose a three-stage framework to improve LLMs' rule-learning ability.
3. The authors conduct experiments on five popular LLMs and compare the results with human's abilities.

**Weaknesses:**

1. The small number of questions in the benchmark may limit its applicability.
2. Improving the output of large language models (LLMs) based on the results of interactions with observations and the environment has become a common approach, as seen in methods like ReACT (https://arxiv.org/pdf/2210.03629) and Reflexion (https://arxiv.org/pdf/2303.11366). The main idea of the IDEA framework is quite similar to these works.
3. The experiment lacks some baseline comparisons, such as using search methods or observation-based reflection approaches.

**Questions:**

What is the difference between the Abduction stage and Induction stage? It seems like these two stage are both about generate / refine a hypothesis based on observations.

---

> ### Author Response · Authors · 2024-11-19
> **Respond to weakness (1)**
>
> Thank you for your comments and suggestions. We would like to address your concerns as follows:
> ### **1, Regarding the limited size of our benchmark**
> - See general response
> ### **2 ReACT/Reflexion and IDEA are solving different tasks**
> - The task we address with RULEARN is fundamentally different from those in ReACT and Reflexion. Their questions, sourced from HotpotQA and ALFWorld, are relatively straightforward, such as:
>     - Provided paragraph…… + Question: Aside from the Apple Remote, what other device can control the program that the Apple Remote was originally designed to interact with? (Hotpot QA)
>     - You are in the middle of a room. Looking quickly around you, you see a cabinet 6, a cabinet 1, a coffee machine 1, a countertop 3, a stove burner 1, and a toaster 1. Your task is to: Put some pepper shaker on the drawer. (ALFWorld)
> - These questions provide all the necessary information or a tool like a search engine that **directly provides all the necessary information**, the agent can solve these tasks without learning and summarizing rules from experiments. Humans would achieve very high accuracy as they are straightforward.
> - In contrast, puzzles in RULEARN are more challenging. Agents do not initially know the "rules" of the environment. For example, in an escape room scenario, an agent needs to figure out a password to open a door. To solve the puzzle, the agent must first determine how the password relates to objects in the room—this requires forming hypotheses and refining them through experimentation. Analogously, ReACT/Reflexion tasks are like open-book exams, while RULEARN requires agents to proactively design experiments to learn/summarize patterns in observations to solve the puzzle. In our experiments, even college-educated humans achieve only a 60% success rate within 15 steps.
> - As you mentioned: “Improving the output of LLMs based on the results of interactions with observations and the environment is a common approach.” Solving puzzles in RULEARN inherently requires an agent to reason and reflect on previous observations. Our baseline approach aligns with ReACT/Reflexion. Building upon ReACT/Reflexion, IDEA enhances this naive reflect/reasoning process by guiding LLM agents to adopt a human-like rule-learning loop that consists of Abduction, Deduction, and Induction to explicitly guide the agents to form and refine the hypothesis. This loop focuses on establishing a general, robust rule that explains all observations in a concise and systematic way. As humans, we do not arrive at a rule by simply reflecting or reasoning shallowly in a single step. Instead, we strategically design a series of experiments that require multi-steps to gather observations, hypothesize rules, and refine the rules—a process that IDEA effectively emulates.

---

> > ### Author Response · Authors · 2024-11-19
> > **Respond to weakness (2)**
> >
> > ### **3, Lack baseline comparison (Our baseline is indeed ReACT and Reflexion)**
> > - Our baseline agent (without IDEA) must reason and reflect on previous experiences to make valid decisions, similar to the approaches used in ReACT and Reflexion. The inspiration for ReACT and Reflexion stems from the idea that, in real-life problem-solving, we need to think before taking action at each step. RULEARN questions, designed to mimic real-life scenarios, naturally require this ability to solve them (as explained in the previous section). Since our baseline follows enssiencially the same process as as ReACT or Reflexion, it could also be named ReACT or Reflexion. We will clarify this in our revisions.
> > - Example of our baseline agent's approach (Output by our baseline agent at each step is summarized; the agent did more reasoning than shown below):
> >    - Step1: Agent select action
> >         - I need to first review all the provided actions. Reflect on Kevin's current situation and goal to assess if each action is logical.
> >             - 1. Input code to the Code Secured Door and try opening it: Kevin doesn't have the code yet.
> >             - 2. Try opening the Code Secured Door with brute force: This is impractical and likely to fail.
> >             - 3. Pass through the Watercolour Gallery Entrance and reach the Watercolour Gallery: This might provide more clues.
> >             - 4. Pass through the Acrylic Painting Gallery Entrance and reaches the Acrylic Painting Gallery: This might provide more clues.
> >         - Choose the most logical action. The note says "Focus on blue it hides the truth." Since Kevin has already checked the blue painting in the Oil Painting Gallery, exploring other galleries might reveal more blue-themed paintings or clues.
> >         - Finally, indicate the selected action by placing its numeral in square brackets.
> >         - [1]
> >         - **Result: Found a new paint "River Reflections Watercolor Paint"**
> >     - Step2: Agent select action
> >         - Kevin finds a new painting now access each action with new clues:
> >             - (Similar to step 1, the agent accesses each available action)
> >         - Finally, Indicate the selected action.
> >         - [1]
> >     - Step3: Agent decides to input the password to the door:
> >         - The note says to focus on blue. Kevin has seen two blue paintings. The password consists of three digits. Given the emphasis on blue, perhaps the digits are derived from these paintings. Since there are two blue paintings, the digits could be the number of blue paintings (2) and the number of words in the titles of these paintings (Midnight Dance = 2 words, River Reflections = 2 words). Thus, the password could be 222.
> >         - Finally: Generate the final answer in square brackets.
> >         - [222]
> >         - **Result: All three digits are incorrect. Warning: two attempts are left before the password resets with a new color.**
> >     - Step4: Agent selects action with new feedback.
> >         - ...
> > - These outputs demonstrate how our baseline agents are required to think critically and sequentially, reflecting on past interactions to solve puzzles in RULEARN. This approach embodies the core idea of ReACT and Reflexion.
> > - We will add this comparison in our revision later.

---

> ### Author Response · Authors · 2024-11-19
> **Respond to weakness (3) and Answering the Question**
>
> ### **Why Search-Based Baselines Are Inapplicable in Our Environment:**
> - Since the agents in RULEARN can input any letter or digit to interact with the environment, resulting in an extremely large action space, with many puzzles even featuring an infinite action space. For example, in the Reactor puzzle, an agent could use the provided rules to combine any letters and write an entire novel. Similarly, in the Function Operator puzzle, the agent can input any integer or supported mathematical expression to test its hypotheses.  Consequently, using  search-based methods would not solve any of the puzzles within the 15-step limit  (puzzles not solved within 15 steps are considered failures). Therefore, we do not include search-based methods in our approach.
> - Search-based methods with large or unbounded action spaces typically require a well-defined heuristic or sampling strategy and often need many steps to solve a task. Designing such heuristics in advance is not feasible in our environment because, since the underlying rules are unknown, it is impossible to define a heuristic function, sampling method, or optimization objective for the search.
> - For example, given an observation in the reactor puzzle where "A + B → AB," the reaction can be explained by several aspect (and one can easily come up with more):
>     - **Direct Combination**: The reaction rule simply combines the letters in the order provided.
>     - **Alphabetical Order**: The reaction rule combines two letters in alphabetical order.
>     - **Closed Spaces in Characters**: The reaction rule is based on the number of closed spaces in the letters (e.g., 'A' has one closed triangle, 'B' has two closed semicircles).
> - Each hypothesis would require a different heuristic or sampling method. Enabling search without resorting to brute force would necessitate the agent dynamically designing a heuristic or sampling strategy each time the hypothesis is refined—a highly challenging problem beyond the scope of a single work.
> - Even if such a dynamic search-based method were developed, our experiments limit the agent to solving a puzzle within 15 steps. Given that each puzzle may have different underlying rules and is designed to prevent brute-force solutions (e.g., parameters in Function Operator puzzles change after four failed attempts, and passwords in Escape Room puzzles change after three failed attempts), search-based methods lacking "human-level" reasoning are unlikely to succeed in RULEARN.
>
> ### **Answer to the Question: Difference between abduction and induction**
> - In the abduction stage, the agent establishes a hypothesis based on initial information. In the induction stage, the agent takes the previous hypothesis as input and attempts to refine it. The input to abduction and induction differs, as induction considers the previous hypothesis with the goal of refining it with new observations.
>     - **Abduction**: Generates a hypothesis to explain all initial observations.(Automatically called one time at the beginning to form an initial hypothesis)
>     - *(Input: initial observations → Output: a new hypothesis)*
>     - **Deduction**: Generates a plan to verify this hypothesis. (Automatically called when the hypothesis is generated or refined)
>     - *(Input: hypothesis, goal → Output: plan)*
>     - **Induction**: Refines the previous hypothesis with new observations. (The agent proactively selects this action from the action space when it detects inconsistencies between the observations and the current hypothesis.)
>     - *(Input: old hypothesis, new observations, old observations → Output: refined hypothesis)*

---

> > ### Comment · Reviewer_fgiR · 2024-11-26
> >
> > Thanks for the responses. I would like to maintain my rating mainly because the following reasons:
> > 1.  As the authors say, their methods are built upon ReACT/Reflexion, enhancing this naive reflect/reasoning process by guiding LLM agents to adopt a human-like rule-learning loop. I think that incorporating a human-like loop using prompts, compared to current work, is not a substantial contribution.
> > 2. I have concerns about the terminology used in the paper, such as "Abduction" and "Deduction." The difference between them seems to depend solely on whether the input includes previously generated hypotheses. I believe using terms like "Generate" and "Refine" might be more intuitive and easier to understand.
> >
> > Overall, I acknowledge the contributions made by the authors.  I believe the paper would be even more impressive if the authors proposed more contributions to the method.

---

### Official Review · Reviewer_6uBM · 2024-11-06

**Soundness:** 3
**Presentation:** 3
**Contribution:** 2
**Rating:** 5
**Confidence:** 5

**Summary:**

This works proposes RULEARN, a benchmark consists of interactive puzzles that are designed to assess the rulelearning abilities of LLM agents in interactive settings. It also proposes IDEA, an agent that dynamically establishing and applying rules that mimic human rule-learning behaviors. It has the following three steps: (1) generates initial hypotheses from limited observations through abduction, (2) devises plans to validate these hypotheses or leverages them to solve problems via deduction, (3) refines previous hypotheses using patterns identified from new observations through induction. With the novel testbed and the rule-learning framework, this paper also compares LLM agent rule-learning abilities with that of humans. Such study shows several interesting insights: (1) LLM agents continue to struggle with efficiently exploring unfamiliar environments. (2) LLMs fall short in deducting valid plans to verify current hypotheses
and guide future exploration. (3) LLMs fall short to correct previous hypotheses when they contradict new observations and are less capable of refining a hypothesis to make it more robust.

**Strengths:**

- The RULEARN benchmark is a novel benchmark for agentic rule-learning abilities of LLMs, complementary to previous resources for assessing LLM induction abilities.
- The IDEA framework shows promises in enabling more enhanced rule-learning ability for LLM-based agents.
- The human study provides very interesting insights comparing human and LM rule induction abilities. It's especially interesting to see humans are not very good at formulating initial hypotheses.

**Weaknesses:**

- There lacks details for how the puzzles in the benchmark are curated. Also, there's not discussion around the quality and diversity of the benchmark.
- The size of the benchmark is relatively small compared to usual benchmarks.
- The paper lacks important details of the human study, e.g., how are human participants are instructed about the task, or how are participants enforced to follow the abduction-deduction-induction framework. These details are critical for accessing the rigor of the human studies.
- Minor typo "evaluat" at L117
- Missing critical citation that thoroughly tested iterative rule refinement ability of LMs: https://arxiv.org/pdf/2310.08559

**Questions:**

- How do you ensure the feedback from the environment is accurate in the feedback stage? How are environment feedback given (i.e., are they static pre-coded rule-based feedback, or are they dynamically provided feedback by another LM)?
- Could you be more specific about how the benchmark is created? How do you verify the quality and diversity of the benchmark?
- Have you tested the statistical significance of the human vs. agent comparison?
- Could you provide more details of how human participants are instructed about the task, and how they're trained at the beginning of the study? It'll be very helpful to provide more details of the exact human experiment setup, by providing concrete details of the procedure of the study.
- Might you have suggestions of how to improve the agentic framework for this task as future directions, based on the insights of this current study?

---

> ### Author Response · Authors · 2024-11-19
> **Respond to weakness**
>
> Thank you for your comments and suggestions. We would like to address your concerns as follows:
>
> ### **1, Regarding the limited size of our benchmark**
> - See general response.
> ### **2, Regarding the details of the benchmark construction, quality, and diversity of the benchmark**
> - Since rule discovery is a broad area without a consensus on the number of task classes, we categorize the rules into three types to simulate real-life scenarios. 1) Tasks with clear systemic knowledge that support rule discovery, like math in function operator puzzles. 2) Tasks without systemic knowledge, where the rules are coarse and rely on simple intuition, such as counting attributes and clustering (e.g., escape room puzzles). 3) Tasks with no systemic knowledge and rules that need to be discovered from more fine-grained patterns.
> - Within each type of puzzle, after defining the rules, we progressively populate the puzzle environment with settings ranging from easy to hard.
>     - For function operator puzzles, we gradually include more parameters and functions, and more complex elementary functions like “sin(x)”, “1/x”, and “x^2”. We found that smaller models can solve puzzles with one function and up to two different parameters, while larger models struggle with puzzles containing more than three functions and five parameters. After establishing the difficulty levels, we filled the puzzle set with varying combinations of 1 to 3 functions and 1 to 6 parameters, totaling 20 different combinations.
>     - In escape room puzzles, the process is similar, but the difficulty ranges from puzzles with 3 paintings to 10, and from 1 room to 3 rooms. We evenly distributed puzzles with different numbers of paintings and rooms.
>     - For reactor puzzles, we defined four different reaction rules, ranging from easiest to hardest. Smaller models could solve puzzles with the simplest rule, larger models were challenged by the third synthesis rule, and only humans could solve some of the puzzles under our fourth synthesis rule. Out of 20 puzzles, there are 5 for each rule, with target strings to synthesize ranging from the shortest "ACB" to the most complex "FEADE".
> - We have a clearer table in Appendix A.2 that specifies the difficulty span of our puzzle set.
> ### **3, Regarding the details of human study**
> - We recruit human participants by sending emails via our CS department mailing list, informing them that they will complete text-based puzzles. Screening is unnecessary as the puzzles in RULEARN only require common human knowledge and high-school level math skills, which are standard for college CS major students. Upon arrival at our lab, participants are presented with a consent form that includes essential information about data safety, as well as their rights and responsibilities.
> - After they acknowledge the consent form, we install and run our execution program on their laptops. Participants then type their responses using a keyboard according to the prompt displayed on the screen. The prompt seen by humans is exactly the same as that for the LLM in IDEA.
> - The only difference between the LLM and human participants is that we emphasize certain details like formatting and input layout before starting the evaluation, which are already mentioned in the prompt. This is necessary because humans tend to skim through prompts rather than reading them as meticulously as LLMs.
> - Each participant solves three independent puzzles, one from each category, ensuring they do not know the rules beforehand. This setup helps maintain the integrity of the evaluations.
> ### **4, Regarding missing citations**
> - Just like the “Hypothesis Search” [1] that we've already cited, the approach mentioned in the paper “PHENOMENAL YET PUZZLING” involves providing LLMs with a fixed set of observation samples, showing some samples initially and gradually revealing the rest for LLMs to refine their hypotheses. However, this approach misses a crucial aspect of real-life rule discovery: we do not merely wait to be fed with carefully prepared observations. Instead, we humans proactively design experiments to decide what patterns we want to observe to establish our theories. We aim to mimic this process in RULEARN, where agents can design experiments to verify their hypotheses using provided fine-grained interactive actions. Performance varies significantly among models; with IDEA, larger models like GPT-4o are capable of designing necessary experiments to reveal patterns, while smaller models tend to repeat meaningless attempts that yield no additional information. This ability to proactively gather information with tools to discover unknown rules is not captured in current benchmarks.
> - We will add the citation of this work in related work with other similar papers.
>
> *[1] Ruocheng Wang et al. Hypothesis Search: Inductive Reasoning with Language Models*

---

> ### Author Response · Authors · 2024-11-19
> **Answering the questions**
>
> ### **Q1, How do you ensure the feedback from the environment is accurate in the feedback stage? How are environment feedback given (i.e., are they static pre-coded rule-based feedback, or are they dynamically provided feedback by another LM)?**
> - Feedback is generated by pre-coded interactive objects within our system, which have well-defined interactive actions that yield deterministic and accurate responses based on the agent's input. For example:
>     - In function operator puzzles where the underlying function is f(x) = 2*x+4x^2. When an agent decides to input a value into this function and selects a number like “1”, RULEARN will parse the input, directly calculate f(1), and return the value 6 as the feedback. This result is then formatted into our memory template as “You input 1 into the function, and the output is 6.”
>     - In Escape Room puzzles, the password is a 3-digit number where each digit represents the count of paintings of a specific type—watercolor, oil, or acrylic—that share a specified color. For example, if there is 1 blue watercolor painting and 2 yellow oil paintings, with the specified color initially set to blue, the ground truth password would be “100” at the start. The system will provide a hint to the agent about the specified color, e.g., “The hint color now is blue”. At each step, the agent dynamically explores objects in the environment and reasons about the password based on all paintings in the scene and the current specified color from the hint. When the agent inputs “102” into the door, a templated memory states: “You input 102 into the door and the password is incorrect. The first and second digits are correct, but the third digit is incorrect. You have two more chances before the hint color changes and the password resets.” After three failed attempts, the system updates: “Now the hint color changes to yellow”. And the ground truth password is dynamically updated to 020, reflecting the two yellow oil paintings in the room. (Note: The first digit represents watercolor paintings, the second digit represents oil paintings, and the third digit represents acrylic paintings; these categories do not change after 3 failed attempts.)
> ### **Q2, Could you be more specific about how the benchmark is created? How do you verify the quality and diversity of the benchmark?**
> - See response to concerns Section 2.
> ### **Q3, Have you tested the statistical significance of the human vs. agent comparison?**
> - The final scores in the benchmark are calculated using a 95% confidence interval. The lower bound of human scores surpasses the higher bound of the best-performing LLM (e.g., GPT-4o), indicating that the performance difference between humans and current LLMs is statistically significant.
> ### **Q4, Could you provide more details of how human participants are instructed about the task and how they're trained at the beginning of the study? It'll be very helpful to provide more details of the exact human experiment setup by providing concrete details of the study's procedure.**
> - See response to concerns Section 3.
> ### **Q5, Might you have suggestions of how to improve the agentic framework for this task as future directions, based on the insights of this current study?**
> - Although IDEA significantly enhances rule-learning abilities across a broad range within RULEARN, we have identified areas for further improvement.
>     - One limitation is IDEA's handling of long contexts. As exploration progresses and more observations are encountered, IDEA must process all observations simultaneously. This constraint limits its effectiveness in scenarios with lengthy contexts and complex rules that require extensive experimentation for discovery. By filtering out less critical observations, we can focus on more valuable observations, thus enhancing IDEA's performance in long-context scenarios and more challenging puzzles that necessitate multiple steps to gather evidence.
>     - The current implementation of IDEA begins with a single hypothesis and continuously refines it. However, it could be more beneficial if the agent maintained multiple hypotheses and dynamically refined and selected the most reliable one. Such a strategy would allow for more diverse approaches to exploring the environment and potentially increase the efficiency of puzzle-solving.

---

> > ### Comment · Reviewer_6uBM · 2024-11-24
> >
> > Thanks for the author's careful response. The rebuttal resolves some of my concerns, while some other of my concerns remain. Overall, I appreciate the efforts the author made in this work, while believe it can benefit from more expansion and refinement in a future iteration of the work. I keep my scores as-is. Below are a few points that I believe are helpful to consider for the next iteration of the work:
> >
> > - While the authors have given compelling reasons for the relatively small dataset size, it remains a concern to me that the puzzles have limited quantity and diversity in configurations, and thus can only draw limited conclusions. Especially, while it's a great practice to run several trials for the same puzzle. With the limited computation budget the authors explains, it might be more valuable to run less trials over the same puzzle but instead running experiments over more puzzles. Ideally, the next iteration of this work can benefit from scaling up and diversifying the evaluation set more.
> >
> > - I appreciate the human study in this work, which reveals interesting insights of comparing LMs vs. humans in the IDEA framework. However, it's slightly unconvincing to me as of why humans need to be forced to adapt the IDEA framework for solving the task---it's might be that humans have completely different puzzle solving process naturally and forcing them to adopt the IDEA framework is thus an artificial setup.
> >
> > - In addition, I'd like to encourage the authors to include necessary details of the experiment setup, the dataset construction, the dataset quality check, and the human studies in the next iteration of the work. Also, the paper can benefit from more indepth discussion of how IDEA framework can inform future directions.

---

### Author Response · Authors · 2024-11-19
**Common concern: Puzzle size is too small (1)**

### **Our puzzle size is statistically sufficient** to distinguish model performances using confidence intervals.
- We calculated LLMs' performance confidence intervals, revealing variability that highlights each model's distinct capabilities, confirming our puzzle number suffices for performance evaluation. For example, GPT-4o's baseline success rate confidence interval is (19.21, 28.79); after applying IDEA, it changes to (30.14, 40.9), giving us 99.91% confidence that IDEA improves GPT-4o's performance. Similarly, we're 97.84% confident for LLama-70B and 99.97% for GPT-3.5. However, we reject the hypothesis that IDEA improves performance for LLama3-8b and Gemma-7B, as IDEA adds context understanding burden for low-capacity models. These results align with our paper's findings.
- Moreover, within the same setting, we're statistically confident that the performance ranking in the IDEA setting is: Human > GPT-4o ≈ LLama3-70B > GPT-3.5-Turbo > LLama3-8B > Gemma-7B. Therefore, our puzzle size suffices to distinguish model performance across settings. Adding more puzzles increases the evaluation burden without adding meaningful information.
### **The puzzle size is financially optimized** given our budget and computational constraints.
- One trial of all puzzles (60 puzzles, expanded to 180 across 3 settings) incurs the following costs per trial:
    - 37.62$ for GPT-4o api call
    - 7.524$ for GPT-3.5 api call
    - Approximately 1 day for Llama3-70B using 8 RTX-A6000 GPUs.
    - Approximately 0.5 days for Llama3-8B and Gemma-7B, also with 8 RTX-A6000 GPUs.
    - 150\\$ for recruiting human participants. (We reduced the number of puzzles tested by humans by half, thus the actual number of puzzles per trial is 90 for human evaluation; otherwise, this cost would double to $300 per trial.)
- Over 5 trials, we spent over $1,000 and utilized 80 GPU days for open-source models. The experimental cost is higher compared to traditional QA datasets because each puzzle requires over 20 steps, with context accumulating rapidly during solving.
### **The puzzle set covers a range of difficulties** from simple to difficult.
- Since rule discovery is a broad area without a consensus on the number of task classes, we categorize the rules into three types to simulate real-life scenarios. 1) Tasks with clear systemic knowledge that support rule discovery, like math in function operator puzzles. 2) Tasks without systemic knowledge, where the rules are coarse and rely on simple intuition, such as counting attributes and clustering (e.g., escape room puzzles). 3) Tasks with no systemic knowledge and rules that need to be discovered from more fine-grained patterns.

- Within each type of puzzle, after defining the rules, we progressively populate the puzzle environment with settings ranging from easy to hard.
    - For function operator puzzles, we gradually include more parameters and functions, and more complex elementary functions like “sin(x)”, “1/x”, and “x^2”. We found that smaller models can solve puzzles with one function and up to two different parameters, while larger models struggle with puzzles containing more than three functions and five parameters. After establishing the difficulty levels, we filled the puzzle set with varying combinations of 1 to 3 functions and 1 to 6 parameters, totaling 20 different combinations.
    - In escape room puzzles, the process is similar, but the difficulty ranges from puzzles with 3 paintings to 10, and from 1 room to 3 rooms. We evenly distributed puzzles with different numbers of paintings and rooms.
    - For reactor puzzles, we defined four different reaction rules, ranging from easiest to hardest. Smaller models could solve puzzles with the simplest rule, larger models were challenged by the third synthesis rule, and only humans could solve some of the puzzles under our fourth synthesis rule. Out of 20 puzzles, there are 5 for each rule, with target strings to synthesize ranging from the shortest "ACB" to the most complex "FEADE".
- We have a clearer table in Appendix A.2 that specifies the difficulty span of our puzzle set.

---

> ### Author Response · Authors · 2024-11-19
> **Common concern: Puzzle size is too small (2)**
>
> ### **Our benchmark induces significantly larger data volumes** compared to traditional benchmarks and the size is comparable to other fine-grained interaction environments.
> - Comparing RULEARN to simpler QA datasets or coarse-grained environments like ALF-World—limited to actions like 'go to', 'open', 'take'—is unfair. Our benchmark offers comprehensive interactive environments per puzzle, featuring fine-grained actions like inputting characters or digits and receiving tailored feedback at the character level. For instance, in the Function Operator puzzle, an agent can input any floating-point number into one of the functions and observe the corresponding output. This level of fine-grained interactivity enables agents to design experiments with unlimited degrees of freedom to test and refine their hypotheses.
> - Environments with fine-grained action spaces, like RULEARN, typically generate larger data volumes than traditional QA datasets or coarse-grained interactive environments. In RULEARN, agents must take proactive steps to gather observations, with each decision influencing the diversity of observations they encounter. This interaction significantly increases complexity, as it involves accumulating large numbers of interaction steps. Consequently, each puzzle in RULEARN generates data volumes that are substantially larger than those from a single example in other benchmarks. A more apt comparison would be between the number of steps an agent takes to solve a puzzle and the number of examples in traditional benchmarks, given that the data volume per step is comparable. In our experiments, we limited interactions to a maximum of 15 steps per puzzle, equating to at least 300 effective steps per puzzle type when considering additional steps for abduction, induction, and deduction that are not included in this count. We also provide detailed performance breakdown and analysis at the step level, which underscores the complex reasoning and adaptability required from agents in RULEARN, making it a more challenging and comprehensive evaluation framework.
> - Similar projects like "FurnMove" [1] enable agents to move objects to different coordinates in space and rotate them, but feature only 30 environments. Another project, “RoboCasa” [2], includes 100 environments, and “Co-ELA” [3] has 44 environments. These fine-grained interaction environments are typically in robotics. RULEARN is the first benchmark enabling language agents to interact at a fine-grained level for reasoning and rule discovery in text-based environments. Common sizes for such environments are usually under 100.
> ### **RULEARN features over 20 puzzles per type.** The puzzle size in the paper results from manually selecting representative puzzles we are able to evaluate.
> - Puzzles in RULEARN are programmatically generated, allowing production and assessment of as many puzzles as our budget permits. RULEARN is a flexible, extensible framework—akin to a game engine—that **supports automatic generation of unlimited customizable puzzles** across types:
>     - **Function Operator puzzles**: Created by providing different sets of mathematical functions expressed in text.
>     - **Escape Room puzzles**: Created by specifying paintings with defined colors, descriptions, and positions.
>     - **Reactor puzzles**: Created by providing different target strings that agents need to synthesize, along with the rules for synthesis.
> - Additionally, RULEARN allows users to define new interactive object classes with just a few lines of code, enhancing its versatility. We have designed tools to dynamically create puzzles in batches.
>
> *[1] Unnat Jain et al. A Cordial Sync: Going Beyond Marginal Policies for Multi-Agent Embodied Tasks*.
>
> *[2] Soroush Nasiriany et al. RoboCasa: Large-Scale Simulation of Everyday Tasks for Generalist Robots*.
>
> *[3] Hongxin Zhang et al. Building Cooperative Embodied Agents Modularly with Large Language Models*.

---

### Note · Authors · 2024-12-13

I have read and agree with the venue's withdrawal policy on behalf of myself and my co-authors.